# Validating indicators for monitoring availability and geographic distribution of emergency obstetric and newborn care (EmoNC) facilities: A study triangulating health system, facility, and geospatial data

Jewel Gausman[1]*, Verónica Pingray[2], Richard Adanu[3], Delia A. B. Bandoh[4], Mabel Berrueta[2], Jeff Blossom[5], Suchandrima Chakraborty[6], Winfred Dotse-Gborgbortsi[7], Ernest Kenu[4], Nizamuddin Khan[6], Ana Langer[1], Carolina Nigri[2], Magdalene A. Odikro[4], Sowmya Ramesh[6], Niranjan Saggurti[6], Paula Vázquez[2,8], Caitlin R. Williams[2,9], R. Rima Jolivet[1]

1 Department of Global Health and Population, Women and Health Initiative, Harvard University T.H. Chan School of Public Health, Boston, Massachusetts, United States of America, 2 Institute for Clinical Effectiveness and Health Policy (Instituto de Efectividad Clínica y Sanitaria (IECS)), Buenos Aires, Argentina, 3 Department of Population, Family, and Reproductive Health, University of Ghana School of Public Health, Accra, Ghana, 4 Department of Epidemiology and Disease Control, University of Ghana School of Public Health, Accra, Ghana, 5 Center for Geographic Analysis, Harvard University T.H. Chan School of Public Health, Boston, Massachusetts, United States of America, 6 Population Council, New Delhi, India, 7 School of Geography and Environmental Science, University of Southampton, Southampton, United Kingdom, 8 Department of Health Science, Kinesiology, and Rehabilitation, Universidad Nacional de La Matanza, Buenos Aires, Argentina, 9 Department of Maternal & and Child Health, Gillings School of Global Public Health, University of North Carolina at Chapel Hill, Chapel Hill, North Carolina, United States of America

* jmg923@mail.harvard.edu

**Data Availability Statement:** All data have been anonymized to ensure compliance with human

## Abstract

Availability of emergency obstetric and newborn care (EmONC) is a strong supply side measure of essential health system capacity that is closely and causally linked to maternal mortality reduction and fundamentally to achieving universal health coverage. The World Health Organization's indicator "Availability of EmONC facilities" was prioritized as a core indicator to prevent maternal death. The indicator focuses on whether there are sufficient emergency care facilities to meet the population need, but not all facilities designated as providing EmONC function as such. This study seeks to validate "Availability of EmONC" by comparing the value of the indicator after accounting for key aspects of facility functionality and an alternative measure of geographic distribution. This study takes place in four subnational geographic areas in Argentina, Ghana, and India using a census of all birthing facilities. Performance of EmONC in the 90 days prior to data collection was assessed by examining facility records. Data were collected on facility operating hours, staffing, and availability of essential medications. Population estimates were generated using ArcGIS software using WorldPop to estimate the total population, and the number of women of reproductive age (WRA), pregnancies and births in the study areas. In addition, we estimated the population within two-hours travel time of an EmONC facility by incorporating data on terrain from Open Street Map. Using these data sources, we calculated and compared the value of the

subject protections and study protocols. The
anonymized data underlying the findings are
deposited here: Jolivet, R Rima; Gausman, Jewel;
Adanu, Richard; Bandoh, Delia; Beruetta, Mabel;
Chakraborty, Suchandrima; Kenu, Ernest; Khan,
Nizamuddin; Odikro, Magdalene; Pingray, Veronica;
Ramesh, Sowmya; Vázquez, Paula; Williams,
Caitlin; Langer, Ana, 2022, "Validation Data for
"Maternal Death Review Coverage"", https://doi.org/
10.7910/DVN/3UEY4I, Harvard Dataverse, V1,
UNF:6:wh1AhrYk9dDrEMsQSRgELQ== [fileUNF].

**Funding:** This work was supported by the Bill and
Melinda Gates Foundation through an award to
RRJ and AL (Improving Maternal Health
Measurement (IMHM) Capacity and Use, grant
number OPP1169546). Funders had no role in
study design, data collection and analysis, decision
to publish, or preparation of the manuscript.

**Competing interests:** The authors have declared
that no competing interests exist.

indicator after incorporating data on facility performance and functionality while varying the reference population used. Further, we compared its value to the proportion of the population within two-hours travel time of an EmONC facility. Included in our study were 34 birthing facilities in Argentina, 51 in Ghana, and 282 in India. Facility performance of basic EmONC (BEmONC) and comprehensive EmONC (CEmONC) signal functions varied considerably. One facility (4.8%) in Ghana and no facility in India designated as BEmONC had performed all seven BEmONC signal functions. In Argentina, three (8.8%) CEmONC-designated facilities performed all nine CEmONC signal functions, all located in Buenos Aires Region V. Four CEmONC-designated facilities in Ghana (57.1%) and the three CEmONC-designated facilities in India (23.1%) evidenced full CEmONC performance. No sub-national study area in Argentina or India reached the target of 5 BEmONC-level facilities per 20,000 births after incorporating facility functionality yet 100% did in Argentina and 50% did in India when considering only facility designation. Demographic differences also accounted for important variation in the indicator's value. In Ghana, the total population in Tolon within 2 hours travel time of a designated EmONC facility was estimated at 99.6%; however, only 91.1% of women of reproductive age were within 2 hours travel time. Comparing the value of the indicator when calculated using different definitions reveals important inconsistencies, resulting in conflicting information about whether the threshold for sufficient coverage is met. This raises important questions related to the indicator's validity. To provide a valid measure of effective coverage of EmONC, the construct for measurement should extend beyond the most narrow definition of availability and account for functionality and geographic accessibility.

## Introduction

From a measurement perspective, the Sustainable Development Goal (SDG) period has seen a marked proliferation in the number of indicators put forward for monitoring health at all levels–clinical, health system, and policy–including in the area of maternal and newborn health. [1,2] Sustainable Development Goal (SDG) 3.1 calls for all countries to reduce their maternal mortality ratio (MMR) by at least two-thirds from their 2015 baseline by 2030 to achieve a global average maternal MMR of fewer than 70 maternal deaths per 100,000 live births. [3] This MMR target emanated from the "Strategies toward Ending Preventable Mortality (EPMM)," the global guidance report outlining targets and priority recommendations for maternal health and survival during the SDG period. [4] Ending preventable maternal mortality requires countries to effectively address all causes of maternal death by ensuring universal health coverage [5–7] that guarantees quality maternal and newborn care [8,9], and timely access to lifesaving interventions through availability of emergency obstetric and neonatal care (EmONC) [10,11].

Fundamental principles for effective monitoring of maternal health published in 2015 and updated in 2020 call for focus and fit, i.e., a minimum core set of tracer measures should be prioritized for monitoring, selected based on relevance to well-defined objectives that are useful to the end user. [12,13] Given the need to reduce measurement burden while optimizing the relevance and utility of core measures selected for monitoring, those with demonstrated validity—i.e. that show evidence that they reliably and accurately represent the construct of interest–should be prioritized for national and global monitoring. [14,15].

EmoNC availability is a strong supply side measure of essential health system capacity, closely and causally linked to maternal mortality reduction. [16] Monitoring the availability of EmoNC is often done by examining a succinct number of key clinical interventions designed to reflect health system capacity to treat the range of major obstetric complications consisting of parenteral antibiotics, anticonvulsants and uterotonics, manual removal of placenta, removal of retained products of conception, newborn resuscitation, assisted vaginal delivery, cesarean sections, and blood transfusions. Collectively, these interventions are known as the EmONC signal functions, which are classified at the basic-level (BEmONC), which includes the seven signal functions except cesarean section and blood transfusion, and at the comprehensive-level (CEmONC), which includes all nine signal functions. [17] Facilities are designated as BEmONC or CEmONC based on their performance of the signal functions [18]. While accurately tracking health system capacity to EmONC is inarguably a core construct of interest for ending preventable maternal mortality, how best to define and measure this construct is subject to uncertainty.

The WHO indicator "Availability of EmONC facilities" [17] was prioritized as a core EPMM indicator for its potential contribution toward achieving EPMM Key Theme #10: "Strengthen health systems to respond to the needs and priorities of women and girls" [19]. From its name, the indicator "Availability of EmONC" is often understood to focus solely on whether emergency care facilities exists in sufficient quantity to meet need (facility density) [20]. However, studies demonstrate that the number of facilities designated as providing EmONC is often not a true reflection of the emergency care services available to meet the needs of the surrounding population, because not all facilities designated as providing EmONC function as such [21,22]. To provide a meaningful measure of the construct intended for monitoring–that essential emergency interventions are available in facilities that demonstrate readiness to deliver them effectively and are distributed so that they are physically accessible to the population at risk for obstetric and neonatal emergencies in countries—the concept of availability in the indicator must be operationalized multidimensionally to capture aspects of facility functionality, readiness, and geographic accessibility.

The Office for the High Commissioner for Human Rights (OHCHR) declared the right to the highest attainable standard of health as an obligation of state duty-bearers in 2000 and stipulated that its realization is dependent on the following conditions: availability, accessibility, acceptability, and quality (known as the AAAQ framework). In the AAAQ framework, availability is defined as there being enough functioning facilities, goods, or services, with the enumeration and nature of these being dependent on factors at country level. Accessibility is defined to include both geographic access and socioeconomic dimensions of access. The definitions for "Acceptability" and "Quality" in the AAAQ framework reference respectful and culturally appropriate care, and scientifically sound and medically appropriate care, respectively, thereby aligning with the two principal dimensions of the WHO standards for improving quality of maternal newborn care in facilities. [23] The AAAQ framework provides a useful a lens through which to explore the multiple dimensions of the construct behind the indicator "Availability of EmONC."

The global guidelines for monitoring emergency obstetric care developed by Columbia Averting Maternal Death and Disability, UNFPA, UNICEF, and WHO and released in 2009 are currently under revision [17]. Evidence is needed to assess the validity of the indicator "Availability of EmONC" to explore how well it captures the complex construct it seeks to measure overall, as well as to evaluate the individual components of the indicator's definition and metadata.

To respond to this gap, this study seeks to validate several dimensions captured within the construct of "Availability of EmONC." The first dimension is whether EmONC services are

available within health facilities. To do this, we examined the demonstrated ability of a facility to deliver all EmONC signal functions, as well as staffing availability and whether services are offered 24 hours per day seven days per week. The second dimension is facility density, which measures whether there is a sufficient number of EmONC facilities to ensure coverage of essential emergency care. The measure of density of EmONC facilities is affected not only by the numerator (the number of functional emergency facilities) but also by the population reflected in the denominator (the population in need of emergency obstetric and neonatal care services). This study aims to examine these first two dimensions of availability on the impact on the value of the indicator if the numerator is redefined to account for facility performance and functionality, and also if the denominator is redefined to reflect different population groups that are considered to be in need (total population, women of reproductive age, and births).

Finally, the third dimension relates to facility distribution and geographic accessibility. This study aims to explore this dimension by estimating the proportion the population within two-hours travel time while taking into account facility performance and functionality. Travel times to EmONC greater than two hours have been associated with higher maternal mortality [24,25] and measuring the percentage women within two hours travel time of an EmONC facility has been recommended as a coverage target [22]. The inability of previous studies that have estimated travel time to EmONC to account for facility performance and functionality has been recognized as a limitation that has likely led to overestimates of access [26].

## Materials and methods

This study utilized data from three different sources: 1) cross-sectional data derived from the health system, 2) primary data collected from facility records, and 3) geospatial population data to explore the underlying construct for measurement of the "Availability of EmONC," as it relates to the indicator's numerator and denominator, in a multi-step process.

### Study setting

This study took place in four subnational geographic areas in Argentina, Ghana, and India. More details on the selection for each study area is available in the study protocol [27]. In Argentina, we included the provinces of Buenos Aires (Region V), Jujuy, La Pampa, and Salta. In Ghana, we included the districts of Bunkpurugu Yunyoo and Tolon in the Northern Regions and Techiman and Sunyani Municipal in the Brong-Ahafo Region. In India, we included the districts of Gonda and Meerut in the state of Uttar Pradesh and Krishnagiri and Thiruvallur in the state of Tamil Nadu.

### Facility selection

In each study setting, we obtained an official list of all public and registered private facilities that provide birth care from the Ministry of Health. All facilities on the list were categorized according to their official government designation as being a BEmONC-designated facility, a CEmONC-designated facility, or as having no EmONC designation. All facilities identified by the Ministry of Health as providing birth care were included in the study. In total, 34 facilities were identified in Argentina, 51 facilities in Ghana, and 282 facilities in India. All eligible facilities participated in the study, leading to a 100% participation rate.

### Facility data

Study staff visited each facility and reviewed the facility's records to assess the performance of the EmONC signal functions. Facility records that were reviewed included maternity, delivery,

general admissions, operating theater, female ward, discharge, drug inventory, and staffing registers.

From these records, researchers extracted data using a predesigned data collection form on the performance of each EmONC signal function. For performance of signal functions, we sought evidence that the facility had performed the following nine EmONC signal functions at least once in the 90 days preceding data collection: 1) administration of parenteral antibiotics, 2) administration of parenteral oxytocics, 3) administration of parenteral anticonvulsants, 4) manual removal of placenta, 5) removal of retained products of conception, 6) assisted vaginal delivery (vacuum or forceps), 7) newborn resuscitation with bag and mask, 8) cesarean delivery, and 9) blood transfusion was reviewed and documented. Signal functions 1–7 are BEmONC signal functions. CEmONC signal functions comprise all nine interventions.

The following data on EmONC facility readiness were also collected: 24/7 care, 24/7 staffing, and availability of essential drugs. We defined 24/7 care as whether the facility was open to provide care 24-hours per day/7 days per week. We defined 24/7 staffing as whether there were for obstetricians/gynecologists, midwives, auxiliary nurse midwives, or nurses available 24-hour/7 day per week. For essential drugs, researchers recorded whether the facility had a drug inventory register available and whether parenteral antibiotics, oxytocin/ergometrine, magnesium sulfate/diazepam, and misoprostol were in stock at the time of data collection.

Finally, data were collected from clinical records on the number and type of recorded cases of obstetric and neonatal emergencies.

Other covariates were also collected from each facility, including facility type (i.e. primary, secondary, or tertiary.), facility governance (public or private sector), and location (rural or urban). Further, we collected geographic information system (GIS) coordinates from each facility to document its precise geo-location. Facility data were collected and managed using REDCap (Research Electronic Data Capture) which is a secure web-based application designed to support data capture for research studies [28].

## Population data

In each study setting, the total population, number of women of reproductive age, and number of births were estimated using geospatial methods. Population estimates were generated using ArcGIS software [29]. For India and Argentina we first obtained the GIS shapefile for each district/province's administrative boundaries from the Database of Global Administrative Areas, version 2.0 [www.gadm.org]. For Ghana, we obtained a district boundary shapefile directly from the Ghana Ministry of Health. Then, we used WorldPop to estimate the specific populations of interest in each study setting. WorldPop is an open access GIS dataset that estimates human populations by drawing on census data, United Nations population estimates, and satellite imagery [30]. Total population estimates were obtained for the year 2020 and were available in raster format at a resolution of 100m while estimates of number of women of reproductive age and births were available only for the year 2015 and at a resolution of 1km. The ArcGIS "zonal statistics to table" function was applied to each population raster, specifying district border shapefiles as the input zones. This produced total population, number of women of reproductive age, and number of birth estimates for all districts. We used the WorldPop top-down constrained population modeling method to accurately identify rural areas, small settlements, and uninhabited areas [30,31].

Last, using the population estimates within each study setting, we estimated the total population, WRA, and births that occurred within two-hours travel time of a designated, fully or partially functioning BeMONC facility by incorporating data on roads, rivers, and lakes from Open Street Map, which is a mapping project involving both professional cartographers and

citizens that map features such as roads, waterways, buildings, places of interest, and more, primarily using satellite imagery and GPS locations. The data were downloaded in GIS shapefile format from the Geofabrik OSM download server [32]. Data on elevation were obtained from the Shuttle Radar Topography Mission (SRTM), which is an international research effort that mapped elevation on a near-global scale at a ground resolution of 90 meters, and for select locations at 30m resolution [33].

## Map background data source

For the maps in Figs 3, 4 and 6 the "Population density" data displayed in the background was accessed from WorldPop (www.worldpop.org) and is available for use under the Creative Commons Attribution 4.0 International License (Argentina: https://hub.worldpop.org/doi/10.5258/SOTON/WP00674, Ghana: https://hub.worldpop.org/doi/10.5258/SOTON/WP00674, India: https://hub.worldpop.org/doi/10.5258/SOTON/WP00674).

## Ethics statement

The Institutional Review Board (IRB) of the Harvard T.H. Chan School of Public Health approved this study on 4 September 2019 (approval ID: IRB19-1086). The research is classified as Level 4 Data using Harvard's Data Security Policy. The study also was approved in Argentina by the Comité de Ética de la Investigación de la Provincia de Jujuy (approval ID not applicable), Comisión Provincial de Investigaciones Biomédicas de la Provincia de Salta (approval ID: 321-284616/2019), Consejo Provincial de Bioética de la Provincia de La Pampa (approval ID not applicable), and Comité de Ética Central de la Provincia de Buenos Aires (approval ID: 2919-2056-2019); in Ghana by the Ghana Health Service Ethical Review Board (approval ID: GHS-ERC022/08/19); and in India by the national population council IRB (approval ID: 889) and local Sigma-IRB (approval ID: 10052/IRB/19-20).

After full board review, the need for informed consent was waived as it was determined our study did not collect any data on human subjects. Only anonymized and generalizable data were collected during retrospective medical chart review.

## Inclusivity in global research

Additional information regarding the ethical, cultural, and scientific considerations specific to inclusivity in global research is included in the (S1 Checklist).

## Analysis

In the first step of our analysis, we tabulated the performance of each of the nine EmONC signal functions in the 90 days prior to data collection among the health facilities included in the study according to their location, level, and their official EmONC designation. As signal function #6 (assisted vaginal delivery) is not routinely performed in some of the study settings, we created an additional category reflecting partial performance at the BEmONC and CEmONC level to identify facilities in which all B/CEmONC signal functions were performed, less signal function #6.

Next, we analyzed facility functionality. Our definition of a functional EmONC facility corresponds to the UNFPA definition which defines a fully functional EmONC facility as one that performed all seven BEmONC or nine CEmONC signal functions over the previous 90-day period as well as whether the facility is open 24/7 (24/7 care). As before, we also expanded our definition of partial functionality to include facilities with 24/7 care and in which all B/CEmONC signal functions, except #6, were performed. We descriptively analyzed the

percentage of fully and partially performing BEmONC-designated facilities as well as the percentage of fully and partially functioning CEmONC-designated facilities at the both the CEmONC-level and BEmONC-level, given that CEmONC facilities are also by definition BEmONC facilities.

While not included in our definition of facility functionality, we also examined several other elements of facility readiness aside from 24/7 care to further explore the validity of the indicator, including 24/7 staffing, availability of essential medicines, and presences of a drug inventory register. We examined these elements of facility EmONC readiness versus facility EmONC designation by calculating the percentage of facilities that had 24/7 care, 24/7 staffing, and essential drugs in stock.

As a final examination of the numerator's indicator, we used chi squared tests to explore whether facility performance of specific signal functions over the previous 90 days was associated with whether the facility had at least one case of an obstetric emergency that would warrant the performance of that specific signal function. As some clinics may not encounter the specific obstetric emergencies that may warrant performance of a given signal function, this enabled us to examine whether lack of performance of a signal function may be explained by case load, rather than a lack of ability to perform that signal function.

Then, we calculated the value of the indicator using different versions of the numerator and the denominator derived from the study results. We calculated the number of designated, fully functional, and partially functional BEmONC- and CEmONC- facilities per 500,000 population and per 20,000 births. We compared the results to the global reference standard that suggests a minimum number of five BEmONC facilities and one CEmONC facility per 500,000 population or 20,000 births, and explored the variation between these different measures across the study areas [16].

Finally, we calculated the percentage of total population, women of reproductive age, and births within 2 hours travel time of designated, fully functioning, and partially functioning B/CEmONC facilities and compare the results to global recommendations that 90% of the population denominator used be within 2 hours of a EmONC facility [22].

Analyses were performed using Stata v14 and graphical displays were created using the R Package *ggplot* [34].

## Results

**Table 1** shows the distribution of the facilities in the study areas across each country included in the study. In Argentina and Ghana, more that 40% of all the facilities providing birth care were located in one study area, with 47.0% in Buenos Aires Region V for Argentina and 43.1% in Sunyani district for Ghana. In India, facilities were more equally distributed across the four study areas. In Argentina, all facilities were either secondary- or tertiary-level facilities, while in Ghana and India, most facilities provided primary-level care. All facilities included in Argentina were urban, while 85.5% of facilities in India were in rural areas. In Argentina, all facilities were designated to provide care at the CEmONC level. In both Ghana and India, a larger percentage of facilities located in the study areas were designated at the BEmONC-level rather than at the CEmONC-level, though in both settings, facilities with no EmONC designation constituted the majority of all facilities (45.1% in Ghana and 71.6% in India).

### Performance of EmONC signal functions

Facility performance of BEmONC and CEmONC signal functions varied considerably across and within study settings. **Fig 1** shows performance of each signal function across all facilities in each country, regardless of a facility's EmONC designation. In general, across all three

**Table 1. Description of facility characteristics.**

|  | Argentina | Ghana | India |
|---|---|---|---|
|  | % (n) | % (n) | % (n) |
| All Facilities | 100.0 (34) | 100 (51) | 100.0 (282) |
|  |  |  |  |
| Setting |  |  |  |
| Buenos Aires/Bukpurugu Yunyoo/Gonda | 47.0 (16) | 15.7 (8) | 30.1 (85) |
| Jujuy/Sunyani/Krishnagiri | 11.8 (4) | 43.1 (22) | 24.1 (68) |
| La Pampa/Techiman/Meerut | 17.7 (6) | 25.5 (13) | 20.6 (58) |
| Salta/Tolon/Thirvallur | 23.5 (8) | 15.7 (8) | 25.3 (71) |
| Facility Type |  |  |  |
| Primary | 0.0 (0) | 84.3 (43) | 92.9 (262) |
| Secondary | 55.9 (19) | 13.7 (7) | 6.1 (17) |
| Tertiary | 44.2 (15) | 2.0 (1) | 1.1 (3) |
| Facility Location |  |  |  |
| Rural | 0.0 (0) | 31.4 (16) | 85.5 (241) |
| Urban | 100 (34) | 68.6 (35) | 14.5 (41) |
| Governance |  |  |  |
| Public | 100 (34) | 78.7 (40) | 100 (282) |
| Non-profit Private | 0.0 (0) | 2.0 (1) | 0.0 (0) |
| For-profit Private | 0.0 (0) | 19.6 (10) | 0.0 (0) |
| Other | 0.0 (0) | 0.0 (0) | 0.0 (0) |
| Designated EmONC Status |  |  |  |
| CEmONC | 100.0 (34) | 13.7 (7) | 4.61 (13) |
| BEmONC | 0.0 (0) | 41.2 (21) | 23.76 (67) |
| No designation | 0.0 (0) | 45.1 (23) | 71.6 (202) |

countries, there was a high degree of variability in the performance of each of the nine signal functions in birthing sites. In all countries, administration of parental oxytocics was the most commonly performed signal function (100% in Argentina, 92.2% in Ghana, and 95.7% in Argentina), and signal function #6 (assisted vaginal delivery) was the least commonly performed (26.5% of facilities in Argentina, 13.7% of facilities in Ghana, and 1.8% of facilities in India). In Ghana, only 25.5% of facilities performed signal function #1 (administration of parenteral antibiotics), though it was among the most commonly performed signal functions in the other two study countries (100% in Argentina and 87.2% in India).

Performance of all B/CEmONC signal functions as well as 24/7 care, also varied considerably among facilities designated as B/CEmONC in each country. **Table 2** shows the proportion of BEmONC-designated facilities in Ghana and India that had performed all seven BEmONC signal functions in the 90 days prior to data collection, all BEmONC signal functions less assisted vaginal delivery (considered as partial performance), and those that did not meet either standard. As shown in the table, the vast majority of BEmONC-designed facilities in Ghana and India were found to be non-performing. In Ghana, only one BEmONC-designated facility (reflecting 4.8% of all BEmONC-designated facilities), located in Bunkpurugu Yunyoo, had performed all seven BEmONC signal functions and no BEmONC-designated facilities evidenced partial performance. In India, no BEmONC-designated facilities had performed all seven BEmONC signal functions, and only 12 (17.9%) had evidenced partial performance. One district, Thiruvallur, did not have any BEmONC-designated facilities that evidenced either complete or partial performance of the seven BEmONC signal functions.

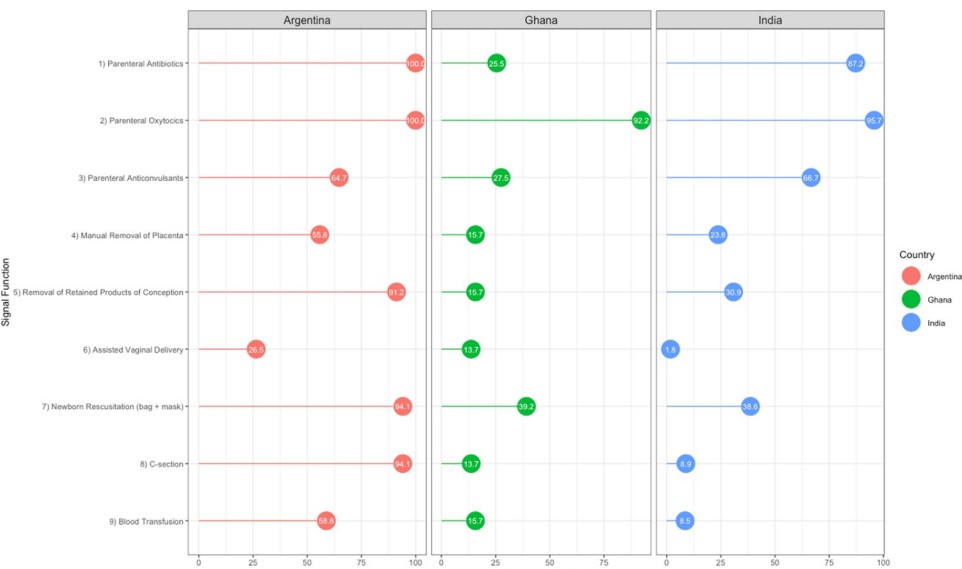

**Fig 1. Performance of signal functions across all birth facilities in study settings in Argentina, Ghana, and India.**

Performance of all seven basic signal functions and the two additional comprehensive signal functions was also fairly limited among CEmONC-designated facilities in the study countries. **Table 3** shows the percentage of CEmONC-designated facilities that performed at either the BEmONC or CEmONC-level, and whether they exhibited full or partial performance. While performance of EmONC signal functions at CEmONC-designated facilities was better than among BEmONC-designated facilities, a substantial proportion of CEmONC-designated facilities did not perform all nine CEmONC signal functions or even all seven BEmONC signal functions in all three countries. In Argentina, only three (8.8%) of all CEmONC-designated facilities performed all nine CEmONC signal functions, and they were all located in Buenos Aires Region V. No additional CEmONC facilities in Argentina evidenced either partial or full performance at the BEmONC-level. Similarly, four CEmONC-designated facilities in Ghana (57.1%) and the three CEmONC-designated facilities in India (23.1%) evidenced full CEmONC performance. As in Argentina, no additional CEmONC-designated facilities evidenced full performance at the BEmONC-level. In all countries but Ghana, removing assisted vaginal delivery from the list of CEmONC functions resulted in an increased number of CEmONC-designated facilities that met the definition for partial performance. In Argentina, eleven CEmONC-designated facilities (32.4%) exhibited partial performance at the CEmONC-level. An additional three facilities (8.9%) exhibited partial performance at the BEmONC-level. In India, an additional four facilities (30.8%) evidenced partial CEmONC performance and one additional facility evidenced (7.7%) evidenced partial BEmONC performance. No CEmONC-designated facilities evidenced partial CEmONC or partial BEmONC performance in Ghana.

A comparison of a facility's performance of a specific signal function during the 90 days prior to data collection to whether a facility had record of a corresponding obstetric emergency during that same time period suggests that in general, there is a significant association between case load and performance, as detailed in **Table 4**. For the most part, facilities that did not perform a specific signal function had no record of encountering an obstetric emergency that would require its performance. For example, among the facilities that failed to perform Signal Function #3 (administration of anticonvulsants), more than 80% did not have a confirmed

**Table 2. Performance of BEmONC functions among BEmONC-designated facilities in Ghana and India.**

| | BEmONC Performance | | | |
|---|---|---|---|---|
| | **All BEmONC\*** | **Partial BEmONC\*\*** | **Non-Performing** | **Total** |
| **Ghana (n = 21)** | | | | |
| All Facilities | 4.76 (1) | 0.0 (0) | 95.2 (20) | 100.0 (21) |
| Setting | | | | |
| Bunkpurugu Yunyoo | 100.0 (1) | 0.0 (0) | 0.0 (0) | 100.0 (1) |
| Sunyani | 0.0 (0) | 0.0 (0) | 100.0 (15) | 100.0 (15) |
| Techiman | 0.0 (0) | 0.0 (0) | 100.0 (4) | 100.0 (4) |
| Tolon | 0.0 (0) | 0.0 (0) | 100.0 (1) | 100.0 (1) |
| Facility Type | | | | |
| Primary | 5.0 (1) | 0.0 (0) | 95.0 (19) | 100.0 (20) |
| Secondary | 0.0 (0) | 0.0 (0) | 100.0 (1) | 100.0 (1) |
| Tertiary | — | — | — | — |
| Facility Location | | | | |
| Rural | 8.3 (1) | 0.0 (0) | 91.7 (11) | 100.0 (12) |
| Urban | 0.0 (0) | 0.0 (0) | 100.0 (9) | 100.0 (9) |
| **India (n = 67)** | | | | |
| All Facilities | 0.0 (0) | 17.9 (12) | 82.1 (55) | 100.0 (67) |
| Setting | | | | |
| Gonda | 0.0 (0) | 26.7 (4) | 73.3 (11) | 100.0 (15) |
| Krishnagiri | 0.0 (0) | 16.7 (3) | 83.3 (15) | 100.0 (18) |
| Meerut | 0.0 (0) | 41.7 (5) | 50.0 (6) | 100.0 (12) |
| Thirvallur | 0.0 (0) | 0.0 (0) | 100.0 (22) | 100.0 (22) |
| Facility Type | | | | |
| Primary | 0.0 (0) | 27.9 (12) | 72.1 (31) | 100.0 (43) |
| Secondary | 0.0 (0) | 0.0 (0) | 100.0 (11) | 100.0 (11) |
| Tertiary | — | — | — | — |
| Facility Location | | | | |
| Rural | 0.0 (0) | 21.2 (11) | 78.8 (41) | 100.0 (52) |
| Urban | 0.0 (0) | 6.7 (1) | 93.33 (14) | 100.0 (15) |

\* "All BEmONC" is defined as performance of all EmONC Signal Functions 1–7 during the 90 days preceding data collection.

\*\*"Partial BEmONC" is defined as performance of all BmONC Signal Functions less assisted vaginal delivery (Signal Function 6) during the 90 days preceding data collection.

case of severe pre-eclampsia in the facility register, which would be a clinical reason to administer anticonvulsants. There are some important exceptions to note. In India, a majority (61.1%) of facilities that did not perform Signal Function #9 (blood transfusion) had at least one case of maternal hemorrhage on file.

## Facility readiness—24/7 care, 24/7 staffing and essential medicine availability by EmONC designation

In general, a majority of birthing facilities in all three countries were open to provide 24/7 emergency care, had at least one staff on call or on duty 24/7 to respond to an obstetric emergency, had all essential drugs in stock (except for non-EmONC designated facilities in Ghana), and had a drug inventory register present on site (**Table 5**). Further, more CEmONC-designated facilities provided 24/7 care and had greater availability of essential medicines than BEmONC-designated facilities or facilities without EmONC designation. Over 90% of B/

**Table 3. Performance of B/CEmONC functions among CEmONC-designated facilities in Argentina, Ghana and India.**

| | CEmONC Performance | | | | BEmONC Performance | | | |
|---|---|---|---|---|---|---|---|---|
| | All CEmONC* | Partial CEmONC** | Non-Performing | Total | All BEmONC* | Partial BEmONC** | Non-Performing | Total |
| **Argentina** | | | | | | | | |
| All Facilities | 8.8 (3) | 32.4 (11) | 58.8 (20) | 100.0 (34) | 8.82 (3) | 41.2 (14) | 50.0 (17) | 100.0 (34) |
| Setting | | | | | | | | |
| Buenos Aires | 18.8 (3) | 25.0 (4) | 56.25 (9) | 100.0 (16) | 18.8 (3) | 31.3 (5) | 50.0 (8) | 100.0 (16) |
| Jujuy | 0.0 (0) | 50.0 (2) | 50.0 (2) | 100.0 (4) | 0.0 (0) | 75.0 (3) | 25.0 (1) | 100.0 (4) |
| La Pampa | 0.0 (0) | 16.7 (1) | 83.3 (5) | 100.0 (6) | 0.0 (0) | 16.7 (1) | 83.4 (5) | 100.0 (6) |
| Salta | 0.0 (0) | 50.0 (4) | 50.0 (4) | 100.0 (8) | 0.0 (0) | 62.5 (5) | 37.5 (3) | 100.0 (8) |
| Facility Type | | | | | | | | |
| Primary | — | — | — | — | — | — | — | — |
| Secondary | 13.3 (2) | 53.3 (8) | 33.3 (5) | 100.0 (15) | 13.3 (2) | 53.3 (8) | 33.3 (5) | 100.0 (15) |
| Tertiary | 5.3 (1) | 15.8 (3) | 79.0 (15) | 100.0 (19) | 5.3 (1) | 31.6 (6) | 63.2 (12) | 100.0 (19) |
| Facility Location | | | | | | | | |
| Rural | — | — | — | — | — | — | — | — |
| Urban | 8.8 (3) | 32.4 (11) | 58.8 (20) | 100.0 (34) | 8.82 (3) | 41.2 (14) | 50.0 (17) | 100.0 (34) |
| **Ghana** | | | | | | | | |
| All Facilities | 57.1 (4) | 0.0 (0) | 42.9 (3) | 100.0 (7) | 57.1 (4) | 0.0 (0) | 42.9 (3) | 100.0 (7) |
| Setting | | | | | | | | |
| Bukpurugu Yunyoo | 100.0 (1) | 0.0 (0) | 0.0 (0) | 100.0 (1) | 100.0 (1) | 0.0 (0) | 0.0 (0) | 100.0 (1) |
| Sunyani | 50.0 (3) | 0.0 (0) | 50.0 (3) | 100.0 (6) | 50.0 (3) | 0.0 (0) | 50.0 (3) | 100.0 (6) |
| Techiman | — | — | — | — | — | — | — | — |
| Tolon | — | — | — | — | — | — | — | — |
| Facility Type | | | | | | | | |
| Primary | 0.0 (0) | 0.0 (0) | 100.0 (2) | 100.0 (2) | 0.0 (0) | 0.0 (0) | 100.0 (2) | 100.0 (2) |
| Secondary | 75.00 (3) | 0.0 (0) | 25.0 (1) | 100.0 (4) | 75.00 (3) | 0.0 (0) | 25.0 (1) | 100.0 (4) |
| Tertiary | 100.0 (1) | 0.0 (0) | 0.00 (0) | 100.0 (1) | 100.0 (1) | 0.0 (0) | 0.00 (0) | 100.0 (1) |
| Facility Location | | | | | | | | |
| Rural | 100.0 (1) | 0.0 (0) | 0.0 (0) | 100.0 (1) | 100.0 (1) | 0.0 (0) | 0.0 (0) | 100.0 (1) |
| Urban | 50.0 (3) | 0.0 (0) | 50.0 (3) | 100.0 (6) | 50.0 (3) | 0.0 (0) | 50.0 (3) | 100.0 (6) |
| **India** | | | | | | | | |
| All Facilities | 23.1 (3) | 30.77 (4) | 38.6 (5) | 100.0 (13) | 23.1 (3) | 38.5 (5) | 38.5 (5) | 100.0 (13) |
| Setting | | | | | | | | |
| Gonda | 50.0 (1) | 0.0 (0) | 50.0 (1) | 100.0 (2) | 50.0 (1) | 0.0 (0) | 50.0 (1) | 100.0 (2) |
| Krishnagiri | 33.3 (1) | 33.3 (1) | 33.3 (1) | 100.0 (3) | 33.3 (1) | 33.3 (1) | 66.7 (2) | 100.0 (3) |
| Meerut | 0.0 (0) | 20.0 (1) | 80.0 (4) | 100.0 (5) | 0.0 (0) | 40.0 (2) | 60.0 (3) | 100.0 (5) |
| Thiravallur | 33.3 (1) | 66.7 (2) | 0.0 (0) | 100.0 (3) | 33.3 (1) | 66.7 (2) | 0.0 (0) | 100.0 (3) |
| Facility Type | | | | | | | | |
| Primary | 0.0 (0) | 25.0 (1) | 75.0 (3) | 100.0 (4) | 0.0 (0) | 0.0 (0) | 100.0 (4) | 100.0 (4) |
| Secondary | 16.7 (1) | 50.0 (3) | 33.3 (2) | 100.0 (6) | 16.7 (1) | 0.0 (0) | 85.3 (5) | 100.0 (6) |
| Tertiary | 66.7 (2) | 0.0 (0) | 33.3 (1) | 100.0 (3) | 66.7 (2) | 0.0 (0) | 33.3 (1) | 100.0 (3) |
| Facility Location | | | | | | | | |
| Rural | 0.0 (0) | 25.0 (1) | 75.0 (3) | 100.0 (4) | 0.0 (0) | 0.0 (0) | 100.0 (4) | 100.0 (4) |
| Urban | 33.3 (3) | 33.3 (3) | 33.3 (3) | 100.0 (9) | 33.3 (3) | 0.0 (0) | 66.7 (6) | 100.0 (9) |

* All BEmONC Signal Functions is defined as performance of Signal Functions 1–7; All CEmONC Signal Functions is defined as performance of Signal Functions 1–9.

**Partial BEmONC and CEmONC is defined as all B/CEmONC Signal Functions less assisted vaginal delivery (Signal Function 6).

**Table 4. Signal function performance versus facility documentation of corresponding obstetric complications in the previous three months.**

| | Argentina | | | Ghana | | | India | | |
|---|---|---|---|---|---|---|---|---|---|
| | % of facilities that performed signal function (n) | | | % of facilities that performed signal function (n) | | | % of facilities that performed signal function (n) | | |
| Signal Function 1 (parenteral antibiotics) * | No | Yes | p-value | No | Yes | p-value | No | Yes | p-value |
| Facility Register Confirmed Case of Sepsis | | | | | | | | | |
| No | | 79.4 (27) | | 97.3 (36) | 69.2 (9) | **0.004** | 100.0 (36) | 40.6 (100) | **<0.001** |
| Yes | — | 20.59 (7) | | 2.7 (1) | 30.8 (4) | | 0.0 (0) | 59.35 (146) | |
| Signal Function 2 (oxytocics) * | | | | | | | | | |
| Facility Register Confirmed Case of Hemorrhage | | | | | | | | | |
| No | | 47.06 (16) | | 100.0 (4) | 0 (0.0) | 0.141 | 100.0 (12) | 35.6 (96) | **<0.001** |
| Yes | — | 52.9 (18) | | 0.0 (0) | 36.17 (17) | | 0.0 (0) | 64.4 (174) | |
| Signal Function 9 (blood transfusion) | | | | | | | | | |
| Facility Register Confirmed Case of Hemorrhage | | | | | | | | | |
| No | 78.5 (11) | 25.0 (5) | **0.002** | 76.2 (32) | 12.5 (1) | **<0.001** | 39.9 (103) | 20.83 (5) | 0.066 |
| Yes | 21.4 (3) | 75.0 (15) | | 23.81 (10) | 87.50 (7) | | 60.1 (155) | 79.2 (19) | |
| Signal Function 3 (anticonvulsants) | | | | | | | | | |
| Facility Register Confirmed Case of Severe Pre-eclampsia/Eclampsia | | | | | | | | | |
| No | 83.3 (10) | 9.1 (2) | **<0.001** | 88.9 (32) | 64.3 (9) | **0.042** | 97.9 (92) | 17.02 (32) | **<0.001** |
| Yes | 16.67 (2) | 90.91 (20) | | 11.1 (4) | 35.7 (5) | | 2.13 (2) | 82.98 (156) | |
| Signal Function 4 (manual removal of placenta) | | | | | | | | | |
| Facility Register Confirmed Case of Retained Placenta | | | | | | | | | |
| No | 100.0 (15) | 10.53 (2) | **<0.001** | 95.35 (41) | 50.0 (4) | **<0.001** | 46.5 (100) | 32.9 (16) | **0.001** |
| Yes | 0.0 (0) | 89.5 (17) | | 4.65 (2) | 50.0 (4) | | 53.49 (115) | 76.12 (51) | |
| Signal Function 6 (assisted vaginal delivery) | | | | | | | | | |
| Facility Register Confirmed Case of Prolonged Labor | | | | | | | | | |
| No | 56.0 (14) | 44.4 (4) | 0.551 | 90.9 (40) | 57.1 (4) | **0.016** | 76.9 (213) | 20.0 (1) | **0.003** |
| Yes | 44.0 (11) | 55.7 (5) | | 9.09 (4) | 42.86 (3) | | 23.1 (64) | 80.0 (4) | |
| Signal Function 8 (cesarean delivery) | | | | | | | | | |
| Facility Register Confirmed Case of Prolonged Labor | | | | | | | | | |
| No | 100.0 (2) | 50.0 (16) | 0.169 | 88.6 (39) | 71.4 (5) | 0.219 | 78.99 (203) | 44.0 (11) | **<0.001** |
| Yes | 0.0 (0) | 50.0 (16) | | 11.36 (5) | 28.57 (2) | | 21.0 (54) | 56.0 (14) | |

*In Argentina, all facilities performed Signal Functions 1 and 2.

CEmONC facilities in Argentina and India provided 24/7 obstetric and neonatal care (91.2% of CEmONC designated facilities in Argentina and 100.0% and 98.51% of CEmONC and BEmONC facilities in India, respectively). In Ghana, however, the pattern was different than that observed in the other countries, with 85.7% of CEmONC-designated facilities and 52.4% of BEmONC-designated providing obstetric and neonatal care 24/7, with a much higher percentage (91.3%) of non-EmONC designated facilities providing 24/7 obstetric and neonatal care. Nearly all B/CEmoNC-designated facilities had at least one staff member on call or duty 24/7, except for one CEmONC-designated facility in Ghana. Furthermore, nearly all

**Table 5. Availability of 24/7 care, staffing, and essential drugs across EmONC facilities in Argentina, Ghana, and India.**

| | Argentina | Ghana | | | India | | |
|---|---|---|---|---|---|---|---|
| | CEmONC | CEmONC) | BEmONC | Non-designated | CEmONC | BEmONC | Non-designated |
| | % (n) | % (n) | % (n) | % (n) | % (n) | % (n) | % (n) |
| Number of Facilities | 34 | 7 | 21 | 23 | 13 | 67 | 202 |
| **Availability of Obstetric and Neonatal Care** | | | | | | | |
| Provides both 24/7 Obstetric and Neonatal Care (% yes) | 91.18 (31) | 85.7 (6) | 52.4 (11) | 91.3 (21) | 100 (13) | 98.51 (66) | 55.94 (113) |
| | | | | | | | |
| **Staffing Availability** | | | | | | | |
| Cadre of staff on call or duty 24h/7d (% yes) | | | | | | | |
| Obstetrician/Gynecologist | 100.00 (34) | 71.43 (5) | 66.67 (14) | 87 (20) | 84.62 (11) | 25.37 (17) | 4.95 (10) |
| Midwife | 91.18 (31) | 85.71 (6) | 80.95 (17) | 91.3 (21) | — | — | — |
| Auxiliary Nurse Midwife | 14.71 (5) | 0.0 (0) | 9.52 (2) | 13 (3) | 61.54 (8) | 62.69 (42) | 40.10 (81) |
| Nurse | 58.82 (20) | 57.14 (4) | 57.14 (12) | 87 (20) | 100 (13) | 95.52 (64) | 49.50 (100) |
| At least one staff on call or duty 24/7 (% yes) | 100.00 (34) | 85.7 (6) | 100.00 (21) | 100.00 (23) | 100.00 (13) | 100.00 (67) | 70.30 (142) |
| | | | | | | | |
| **Availability of Essential Drugs** | | | | | | | |
| Drugs in Stock (% yes) | | | | | | | |
| Parenteral Antibiotics | 100.00 (34) | 100 (7) | 33.33 (7) | 21.7 (5) | 100.00 (13) | 97.01 (65) | 65.35 (132) |
| Oxytocin/Ergometrine | 100.00 (34) | 100 (7) | 80.95 (17) | 65.2 (15) | 100.00 (13) | 98.51 (66) | 73.76 (149) |
| Magnesium Sulfate/Diazepam | 100.00 (34) | 100 (7) | 66.67 (14) | 39.1 (9) | 100.00 (13) | 94.03 (63) | 57.43 (116) |
| Misoprostol | 94.12 (32) | 100 (7) | 52.38 (11) | 17.4 (4) | 92.31 (12) | 83.59 (56) | 53.47 (108) |
| Drug Inventory Register Present (% yes) | 100.00 (34) | 100.00 (7) | 80.95 (17) | 65.2 (15) | 92.3 (12) | 100.00 (67) | 89.60 (181) |

CeMONC-designated facilities had all the essential drugs examined in stock (though two CEmONC facilities in Argentina and one in India did not have misoprostol in stock). In India, the vast majority of BEmONC-designated facilities had essential drugs in stock (ranging from 83.6% having misoprostol in stock to 98.5% having oxytocin/ergometrine in stock). In Ghana, drug availability was much lower in BEmONC-designated facilities. For example, only 33.3% of BEmONC-designated facilities had parenteral antibiotics in stock and 52.4% had oxytocin/ergometrine in stock. Availability of essential drugs was lowest in both India and Ghana among non-EmONC designated facilities.

## Facility functionality (performance and 24/7 care)

Fig 2 presents the percentage of facilities in each study setting according to their highest level of validated EmONC functionality, which combines performance of the seven BEmONC and nine CEmONC signal functions, as well as being open 24/7 to provide obstetric and newborn care. Further, the results are disaggregated by facility location (rural/urban) and level (primary, secondary, and tertiary) to explore disparities by these characteristics. Across all countries, the vast majority of BEmONC-designated facilities, as well as facilities without EmONC designation, were non-functional. While in general, a larger proportion of CEmONC-designated displayed some degree of functionality, an important percentage were non-functional. In Ghana, only 4.8% of all BEmONC-designated facilities were fully functional, and three districts did not have either a fully or partially functional BEmONC facility. In India, 17.9% of all BEmONC-designated facilities were partially functional, and the percentage of facilities with partial functionality varied considerably across the study districts, ranging from 0% of facilities in Thiruvallur to 41.7% of facilities in Meerut being partially functional. Among CEmONC facilities, in all study countries, adding the requirement that a facility be open to provide 24/7

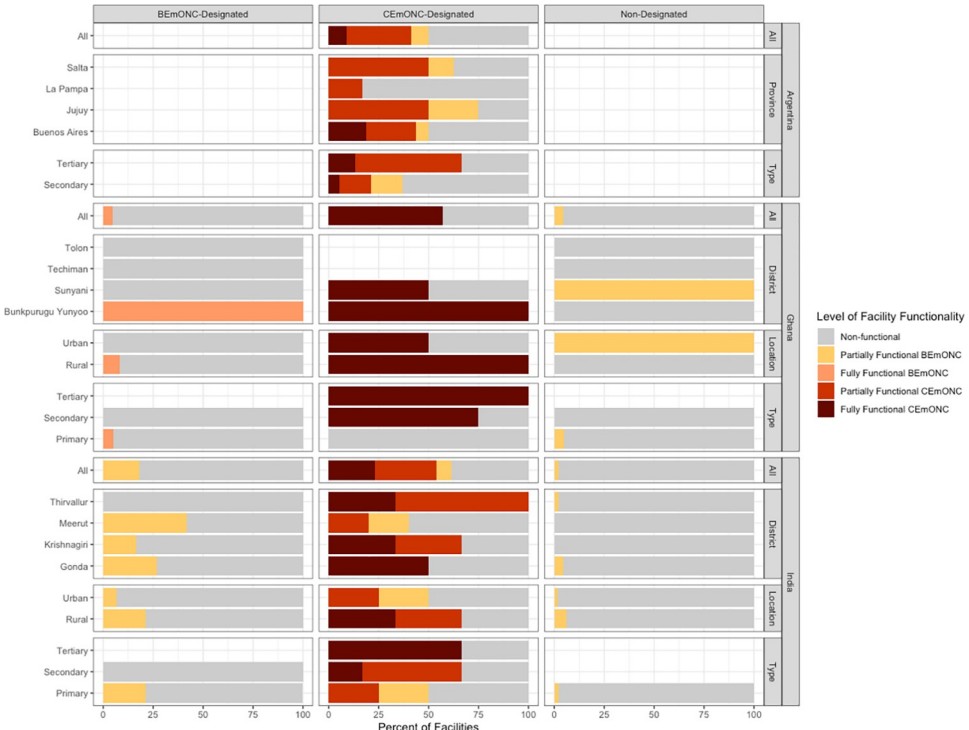

**Fig 2. Percentage of facilities according to their highest level of validated EmONC functionality according to facility characteristics.**

care to be considered fully or partially functional did not reduce the number of facilities operating at either level from those that performed the corresponding signal functions.

## Facility density: Accounting for facility designation and functionality while varying the indicator's reference population

After having examined the different dimensions that comprise the standard and alternative indicator definitions, we next calculated the number of facilities per 500,000 total population and 20,000 births, while accounting for facility designation and functionality (**Table 6**). First, when calculating the indicator to include all EmONC-designated facilities in the numerator regardless of their functionality, a greater number of study areas reached the target of 5 BEmONC facilities per 20,000 births than for 500,000 total population. The magnitude of the difference between indicator estimates obtained based on total population versus births was uneven across study areas within and across countries. Among the three countries in the study, there was the greatest variability across sub-national study areas within India. In Thiruvallur, results showed a 250% increase in the value of the indicator when calculating the indicator per 500,000 total population versus 20,000 births. In Krishnagiri, the magnitude of the difference between the two estimates was slightly lower than in Thiruvallur, yet still exhibited a 150% increase; however, in Gonda and Meerut, there was only a 63% and 73% increase in the value of the indicator, respectively. In Argentina, the difference in magnitude ranged from 100% in Salta to 155% in La Pampa. In Ghana, there was a much smaller magnitude of change in the value of the indicator as calculated based on total population versus births. In Sunyani, the indicator increased by about 170% when calculating it based on total population versus births; however, in Tolon and Techiman, the value of the indicator decreased by about 30%.

**Table 6. Number of EmONC and CEmONC facilities per 500,000 total population and 20,000 births in Argentina, Ghana, and India.**

| Country | Population Estimates | | EmONC Facilities | | | | | | CEmONC Facilities | | | | | |
|---|---|---|---|---|---|---|---|---|---|---|---|---|---|---|
| | | | Designated* | | Partially Functioning | | Fully Functioning | | Designated* | | Partially Functioning | | Fully Functioning | |
| | Total Population | Total Births | Per 500,000 Population | Per 20,000 Births | Per 500,000 Population | Per 20,000 Births | Per 500,000 Population | Per 20,000 Births | Per 500,000 Population | Per 20,000 Births | Per 500,000 Population | Per 20,000 Births | Per 500,000 Population | Per 20,000 Births |
| **Argentina** | | | | | | | | | | | | | | |
| Buenos Aires | 3,252,676 | 53,364 | 2.46 | 6.00 | 1.23 | 3.00 | 1.08 | 1.12 | 2.46 | 6.00 | 1.08 | 2.62 | 0.46 | 1.12 |
| Jujuy | 769,126 | 13,321 | 2.60 | 6.01 | 1.95 | 4.50 | 0.00 | 0.00 | 2.60 | 6.01 | 1.30 | 3.00 | 0.00 | 0.00 |
| La Pampa | 311,725 | 4,877 | 9.62 | 24.61 | 1.60 | 4.10 | 0.00 | 0.00 | 9.62 | 24.61 | 1.60 | 4.10 | 0.00 | 0.00 |
| Salta | 1,336,252 | 26,679 | 2.99 | 6.00 | 1.87 | 3.75 | 0.00 | 0.00 | 2.99 | 6.00 | 1.50 | 3.00 | 0.00 | 0.00 |
| | | | | | | | | | | | | | | |
| **Ghana** | | | | | | | | | | | | | | |
| Bunkpurugu Yunyoo | 186,718 | 6,352 | 5.36 | 6.30 | 5.36 | 6.30 | 2.68 | 3.15 | 2.68 | 3.15 | 2.68 | 3.15 | 2.68 | 3.15 |
| Sunyani Municipal | 135,426 | 3,142 | 77.53 | 133.69 | 14.77 | 25.46 | 11.08 | 19.10 | 22.15 | 38.20 | 11.08 | 19.10 | 11.08 | 19.10 |
| Techiman North | 70,548 | 4,010 | 28.35 | 19.95 | 0.00 | 0.00 | 0.00 | 0.00 | 0.00 | 0.00 | 0.00 | 0.00 | 0.00 | 0.00 |
| Tolon | 52,040 | 2,901 | 9.61 | 6.89 | 0.00 | 0.00 | 0.00 | 0.00 | 0.00 | 0.00 | 0.00 | 0.00 | 0.00 | 0.00 |
| **India** | | | | | | | | | | | | | | |
| Gonda | 3,388,142 | 83,078 | 2.51 | 4.09 | 1.18 | 1.93 | 0.15 | 0.24 | 0.30 | 0.48 | 0.15 | 0.24 | 0.15 | 0.24 |
| Krishnagiri | 1,979,808 | 31,353 | 5.30 | 13.40 | 1.26 | 3.19 | 0.51 | 0.64 | 0.76 | 1.91 | 0.51 | 1.28 | 0.25 | 0.64 |
| Meerut | 3,192,201 | 73,889 | 2.66 | 4.60 | 1.10 | 1.89 | 0.16 | 0.00 | 0.78 | 1.35 | 0.16 | 0.27 | 0.00 | 0.00 |
| Thiruvallur | 3,896,148 | 44,491 | 3.21 | 11.24 | 0.51 | 1.80 | 0.38 | 0.45 | 0.38 | 1.35 | 0.38 | 1.35 | 0.13 | 0.45 |

Coverage estimate exceeds target of five EmONC facilities or one CeMONC facility per 500,000 population or 20,000 births

Next, we explore how the value of the indicator that was calculated using different population parameters is affected when facility functionality is considered (Table 6). Taking into account only fully or partially functioning facilities at the BEmONC-level, the value of the indicator changed dramatically in all countries and sub-national study areas. No sub-national study area in Argentina or India reached the target of 5 BEmONC-level facilities per 500,000 total population or 20,000 births when including only fully or partially functioning BEmONC-level facilities in the numerator. In Ghana, Bunkpurugu Yunyoo and Sunyani reached the target of 5 facilities per 500,000 total population and 20,000 births when partially functional BEmONC facilities were included in the numerator, but only Sunyani reached the target facility density for both total population and births when only fully functioning BEmONC facilities were counted.

At the CEmONC-level, there was less variation across sub-national study areas when changing the value of the numerator to incorporate facility functionality than at the BEmONC-level. All provinces in Argentina reach or exceed the target of one CEmONC facility per 500,000 total population or 20,000 births when considering designated facilities or partially functional facilities; however, only one province (Buenos Aires Region V) reached the target when considering only fully functional facilities. In Ghana, as all CEmONC-designated facilities are fully functional, the value of the indicator did not vary based on the changing definition of the numerator. In India, Krishnigiri, Meerut, and Thiruvallur reached the target of one CEmONC facility per 20,000 births, but not per 500,000 total population, when including CEmONC-designated facilities in the numerator, but Krishnigiri alone reached the target of one CEmONC facility per 20,000 births when only partially functional CEmONC facilities were included in the numerator. No district in India reached the target based on total population or births when only fully functional CEmONC facilities are included in the numerator. In

Ghana, two districts did not have any CEmONC-designated facilities, and no facility of any other designation evidenced full or partial CEmONC performance.

## Percentage of the reference population within two-hours travel time of a designated, partially-, or fully functional EmONC facility

The percentage of the total population, women of reproductive age, and births occurring within two hours travel time of a BEmONC-designated, partially functioning BEmONC, or fully functioning BEmONC facility is presented in Table 7. Changing the population parameter between total population, women of reproductive age, and births, produces little variation across sub-national study areas, and in most areas, does not affect whether the coverage target of 90% of the population within two-hours travel-time is met. However, there are some notable coverage reductions observed when changing the population parameters. For example, the percentage of the total population in Tolon within 2 hours travel time of a designated EmONC facility is estimated at 99.6%; however, only 91.1% of women of reproductive age are within 2 hours travel time–amounting to a reduction of almost 10 percentage points. When factoring in facility functionality, substantial reductions were observed in Argentina and Ghana in the percentage of the population within two-hours travel time from partially and fully functional EmONC facilities. Fig 3 graphically displays the catchment areas within two-hours travel time of a partially functional facility on district maps in relation to population density, and Fig 4 shows the same for fully functional facilities. According to the figures, the location of partially or fully functional facilities does not necessarily correlate with areas have the highest population density within a district, and further, in many districts, even if there is only one partially or fully functional EmONC facility, the majority of the population remains within two-hours travel time to the facility. For example, in Krishnagiri in India, there is only one fully functional EmONC facility (Fig 4), and four partially functional facilities (Fig 3); however, in both cases, 100% of the district is within two-hours travel time.

**Table 7. Proportion of the total population, women of reproductive age and births within 2-hours travel time of designated, partially functional and fully functional EmONC facilities in Argentina, Ghana and India.**

| | Designated EmONC | | | Partially Functional EmONC | | | Fully Functional EmONC | | |
|---|---|---|---|---|---|---|---|---|---|
| | Total Population | WRA* | Births | Total Population | WRA* | Births | Total Population | WRA* | Births |
| Argentina | | | | | | | | | |
| Buenos Aires | 100.0 | 100.0 | 100.0 | 100.0 | 100.0 | 100.0 | 100.0 | 100.0 | 100.0 |
| Jujuy | 97.1 | 96.0 | 96.9 | 97.0 | 96.8 | 95.9 | 0.0 | 0.0 | 0.0 |
| La Pampa | 93.0 | 94.7 | 93.3 | 87.6 | 88.3 | 89.1 | 0.0 | 0.0 | 0.0 |
| Salta | 90.7 | 88.8 | 91.7 | 89.5 | 90.7 | 87.3 | 0.0 | 0.0 | 0.0 |
| Ghana | | | | | | | | | |
| Sunyani | 100.0 | 100.0 | 100.0 | 100.0 | 100.0 | 100.0 | 100.0 | 100.0 | 100.0 |
| Techiman | 100.0 | 100.0 | 100.0 | 0.0 | 0.0 | 0.0 | 0.0 | 0.0 | 0.0 |
| Bunkpurugu Yunyoo | 99.8 | 99.3 | 99.9 | 99.8 | 99.9 | 99.8 | 99.8 | 99.9 | 99.8 |
| Tolon | 99.6 | 91.1 | 99.7 | 0.0 | 0.0 | 0.0 | 0.0 | 0.0 | 0.0 |
| India | | | | | | | | | |
| Thiruvallur | 100.0 | 100.0 | 100.0 | 100.0 | 100.0 | 100.0 | 100.0 | 100.0 | 100.0 |
| Gonda | 100.0 | 100.0 | 100.0 | 100.0 | 100.0 | 100.0 | 100.0 | 100.0 | 100.0 |
| Krishnagiri | 100.0 | 100.0 | 100.0 | 100.0 | 100.0 | 100.0 | 100.0 | 100.0 | 100.0 |
| Meerut | 96.2 | 100.0 | 96.4 | 96.2 | 100.0 | 96.4 | 0.0 | 0.0 | 0.0 |
| *Women of reproductive age | | | | | | | | | |
| | Coverage estimate exceeds target of 90% of population within 2 hours travel time of EmONC | | | | | | | | |

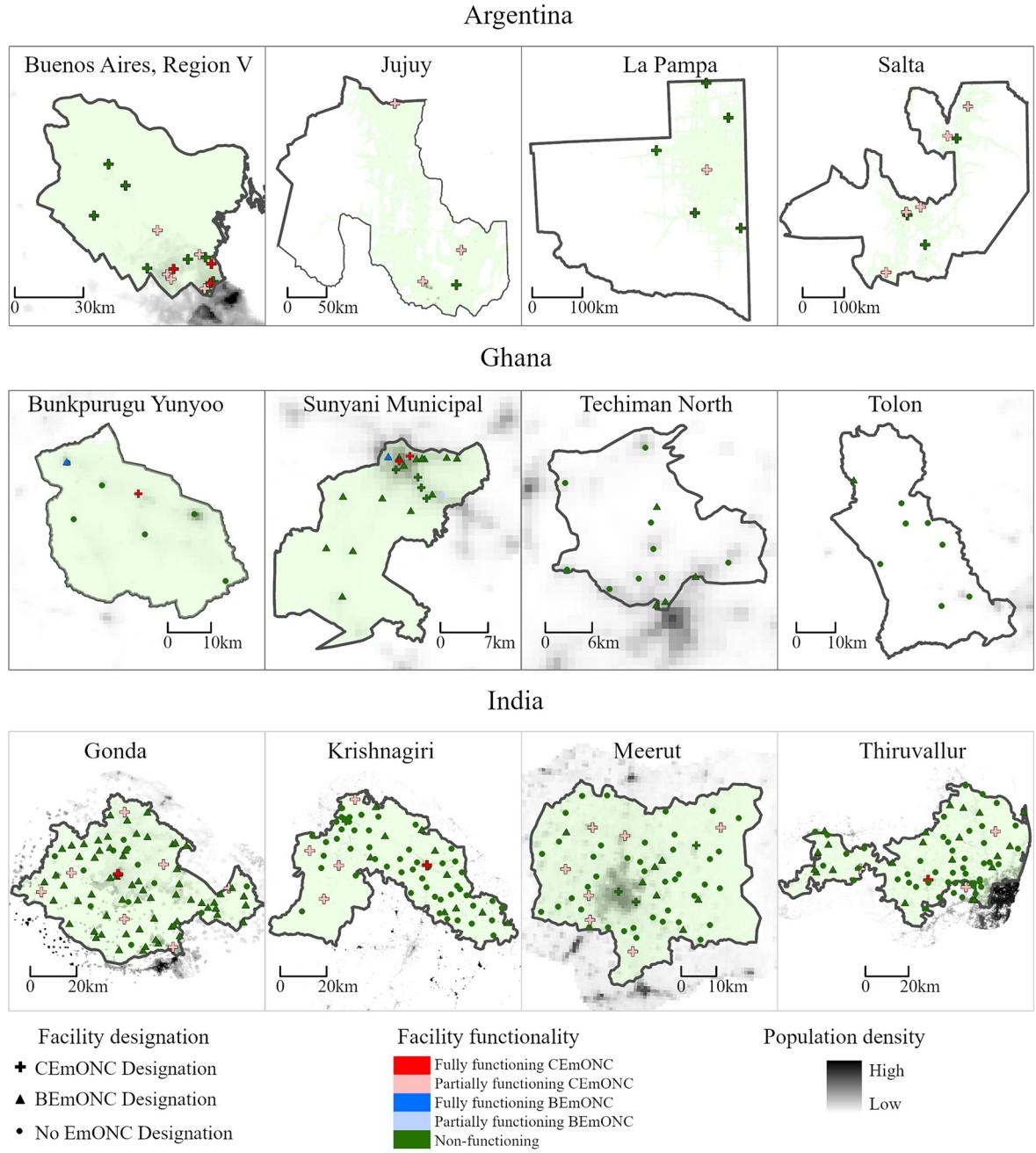

**Fig 3. Geographic distribution of population within two hours travel time of a partially functional EmONC facility.**

**Fig 5** illustrates convergence of the value of indicator measuring the density and distribution of EmONC facilities using the two different estimation approaches: (1) the number of EmONC facilities per 500,000 total population and 20,000 births and (2) the percent of the total population and percent of births within two-hours travel time, taking into account facility designation and functionality using Salta Province in Argentina as an example. In general, changing the numerator or denominator leads to indicator estimates that exceed the suggested targets in one domain but not the other, thus leading to conflicting interpretations of

## Argentina

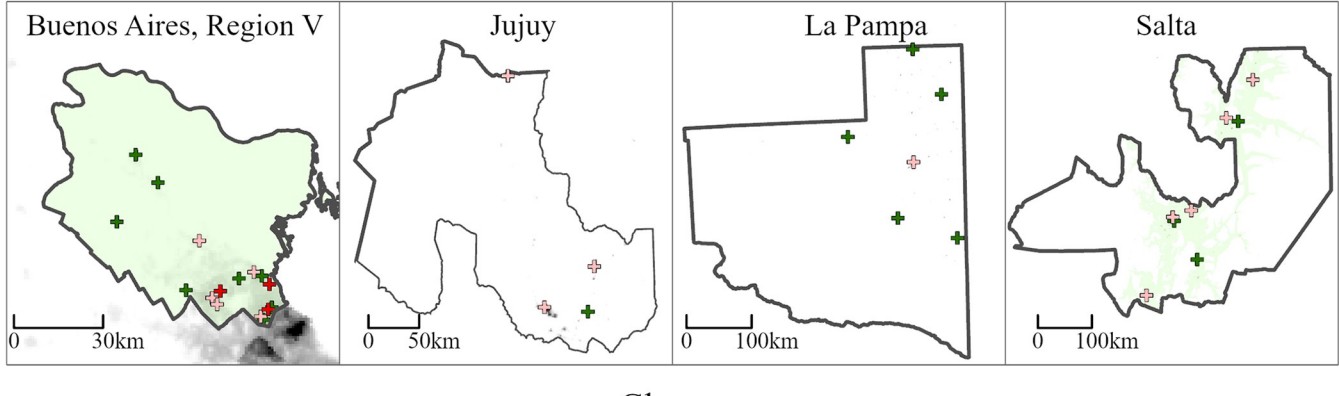

## Ghana

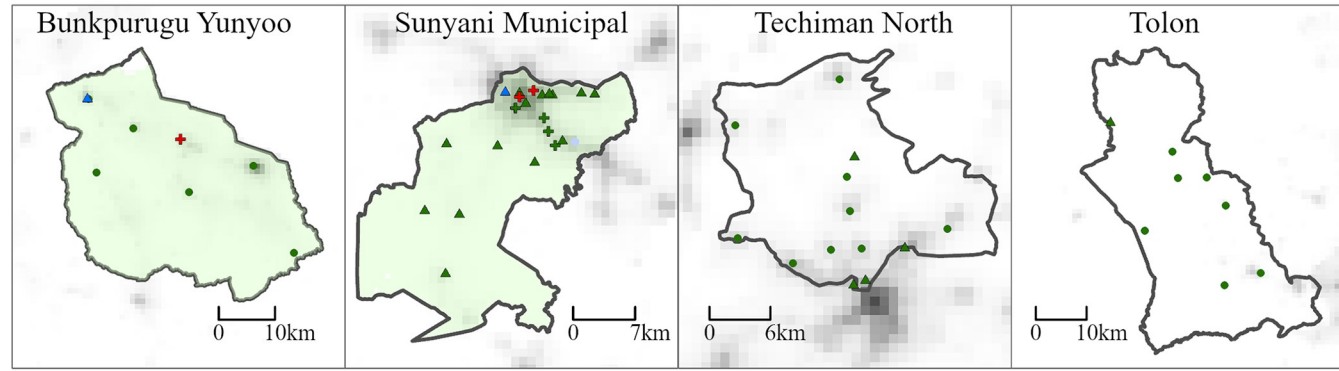

## India

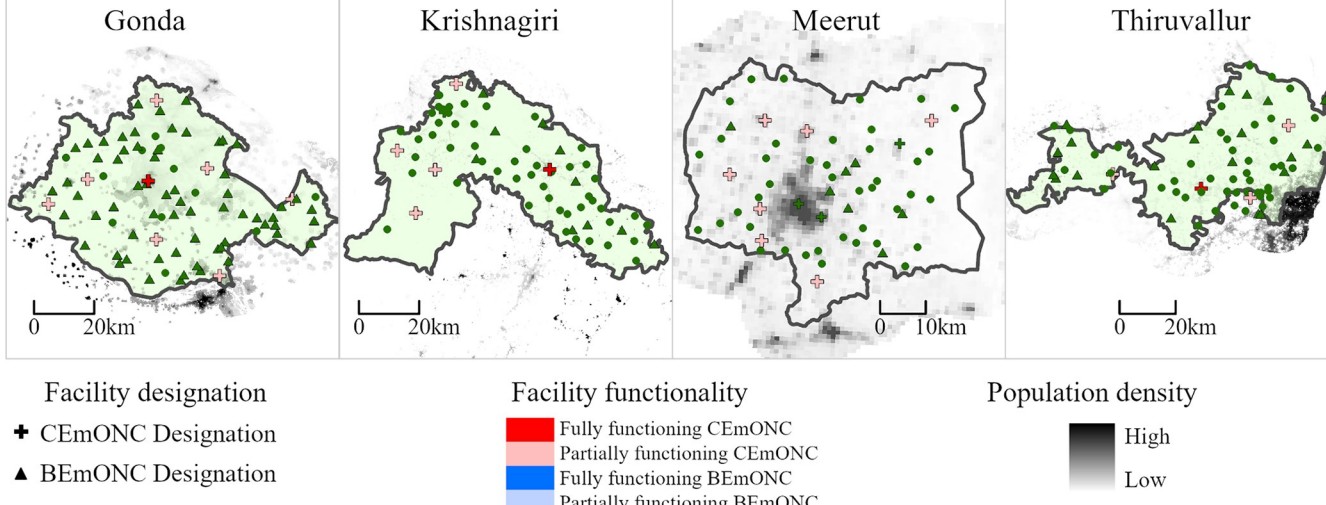

**Facility designation**

✚ CEmONC Designation

▲ BEmONC Designation

● No EmONC Designation

**Facility functionality**

- Fully functioning CEmONC
- Partially functioning CEmONC
- Fully functioning BEmONC
- Partially functioning BEmONC
- Non-functioning

**Population density**

High

Low

Two hour travel catchment area to a district facility demonstrating full functionality

**Fig 4. Geographic distribution of population within two hours travel time of a fully functional EmONC facility.**

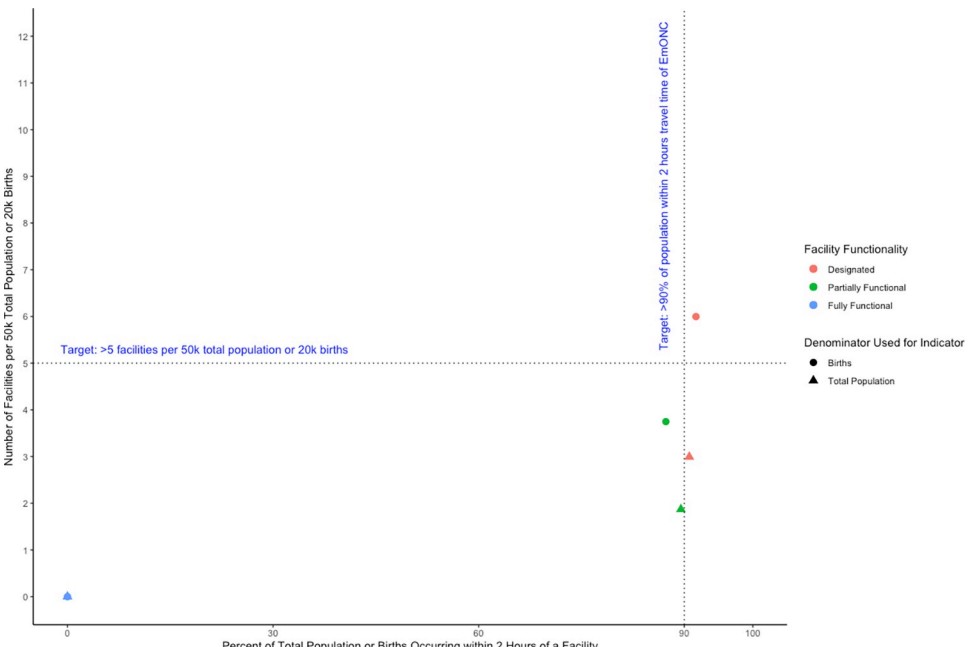

**Fig 5. Comparison of indicators calculated based on facility functionality for Salta Province, Argentina—number of BEmONC facilities per 50k total population, BEmONC facilities per 20k births, percentage of the total population within 2 hours of a BEmONC facility, or percentage of births occurring within 2 hours of a BEmONC facility.**

adequacy. In Salta, the number of EmONC-designated facilities per 20,000 births exceeds the coverage target of five facilities per 20,000 births, and greater than 90% of births occur within two-hours travel time of an EmONC designated facility—again, exceeding the travel-time target. Conversely, there is discordance in whether the value of these two indicators represents sufficient coverage when calculated using the total population as the reference, instead of births. In Salta, there are only three EmONC designated facilities per 500,000 total population, which is far below the recommended number of five; however, at the same time, the percentage of the total population within two-hours travel time of an EmONC-designated facility exceeds the 90% target. When considering partially-functioning facilities with births as the reference population, neither method of calculating the indicator meets either target; however, the percentage of the total population within two-hours travel time of a partially functional facility is only slightly less than the 90% target, but the number of partially functional facilities (1.87) per 500,000 population is substantially less than the target of five.

## Discussion

Our study posits that the indicator "Availability of EmONC," despite the focus implied by its name, is intended to capture a broader construct for measurement that comprises multiple dimensions of availability: availability of all EmONC signal functions within designated EmONC facilities, facility readiness to deliver those essential interventions, coverage of sufficient EmONC facilities to meet the needs of the population, and appropriate geographic distribution of facilities to make those services available in places where they are accessible to those who need them; however, our results question the validity of the indicator as currently defined in incorporating these critical dimensions. Finally, we find that comparing the value of the

indicators when calculated using different definitions reveals important inconsistencies, resulting in conflicting information about whether the threshold for sufficient coverage is met.

Following the AAAQ framework, the indicator explores the dimension of availability as it relates to facility density. Determining whether the quantity of emergency services is sufficient is the ultimate objective for measurement. Sufficiency is defined as the quantity that is "enough to meet the needs of a situation of a proposed end." [35] Since achieving sufficiency is conditional on the best approximation of the need that must be filled, the population used as the denominator to calculate the value of the estimate is significant. While there is no consensus on the optimal parameter to estimate the population in need of EmONC, calculating the indicator using total population versus births reveals considerable differences in magnitude for the estimate of facility coverage, highlighting a lack of convergence between these two measures. Specifically, we find that study areas that achieve sufficiency when it is defined as 5 facilities per 20,000 births my not achieve sufficiency when it is defined as 5 facilities per 500,000 total population. Therefore, establishing a threshold for sufficient coverage is sensitive to demographic characteristics. For example, all the provinces in Argentina and two districts in the state of Tamil Nadu in India have relatively low birth rates. As a result, these areas all meet the threshold of sufficient EmONC density when defining it based on the number of facilities per births, while only one of these areas meets that threshold when defined as facilities per total population. Conversely, we see stronger convergence in the value of the indicator in areas where fertility is higher, such as in Ghana and Uttar Pradesh in India. To date, there is no clear evidence available that identifies the best denominator to use to measure whether there is a sufficient quantity of EmONC facilities.

The evidence base for the target values for sufficient coverage also needs strengthening, since the indicator for unmet obstetric need has not correlated reliably with maternal mortality related to absolute medical indications for emergency interventions. [26] Research has shown that targeting the number of facilities needed per 500,000 population does not correlate well with maternal mortality and instead, targeting the number of facilities needed per births in the population correlates better [16]; however, other plausible denominators could also be tested for their correlation with mortality reduction. The ideal measure of availability of EmONC would be grounded in strong evidence of the population need for emergency care based on the epidemiology of obstetric emergency conditions. Since the minimum thresholds of prevalence for absolute medical indications used to calculate the unmet obstetric need indicator that provides the epidemiologic rationale for the EmONC facility coverage targets may not be valid in all populations, lower rates of surgery for absolute medical indications do not necessarily signal higher maternal mortality from such conditions. [26] In turn, quality and appropriateness of care are potentially influenced by volume of care. In facilities that experience low volumes of emergencies, the risk arises of poorer quality of emergency responses and thus, poorer outcomes of emergency care [27]; however, when all low-risk women are funneled into emergency care settings, the risk of over-intervention increases [28].

While the definition of sufficiency focuses on there being enough supply, it also implies avoiding oversupply. More efficient population coverage of emergency care can be achieved by reducing the number of designated EmONC facilities and instead, focusing on functionality and facility readiness in a smaller number of hospitals [36]. Such an arrangement could have the added advantage of ensuring that the staff in facilities with higher volumes of high-risk cases retains the specialized skills necessary to perform emergency signal functions [37]. This raises questions about the validity of facility density as a measure of EmONC availability, if evidence suggests that more is not necessarily better. To wit, our findings showed that most facilities that perform specific signal functions had no evidence of corresponding complications in their facility records. The reasons for such non-performance are not elucidated by our

research. While it is possible that facilities with poor record keeping failed to record both signal function performance and the corresponding cases of obstetric emergencies, even though both took place, it could also be related to a lack of need to perform those signal functions, which has been cited in other studies as a reason for nonperformance. [17].

Next, looking at "Accessibility," the proportion of the population within two-hours travel time from an EmONC facility has been proposed as an improved measure of coverage over density, as it comprises the element of geographic accessibility. Defining the indicator in this way may be particularly important for determining optimal geographic distribution of facilities for sufficient population coverage, since accessibility is dependent on conditions on the ground, such as terrain, roadways, waterways and climactic factors. [38] In effect, our results demonstrate that greatly reducing the number of facilities by counting only those that are functional in our two-hour travel time calculations only minimally changes the value of coverage estimates in some settings. In a geographically small district such as Krishnagiri in India, for example, 100% of the population is covered within two-hours travel time of a fully functional emergency facility, even though only a small fraction of facilities meet that performance threshold. This supports the argument that effective coverage of emergency care could be achieved with fewer EmONC facilities, entailing more effective use of resources, as long as the functional facilities are distributed geographically so they are accessible to the population in need of them.

The two indicators included in the WHO Manual currently under revision, the first being EmONC facilities per population/births and the second being the proportion of the population with geographic access to EmoNC within two hours travel time, together are meant to fully capture the construct of sufficient supply of facility-based EmONC. Two-hour travel time is only a valid measure of coverage if all emergency facilities demonstrate the same level of functionality. In reality, women do not know which of the facilities within two hours of them are fully-functional, and they will likely seek services closest to them or use other information available to them on quality of care to influence their choice [39]. For health care decision makers concerned with the organization and delivery of obstetric and neonatal care, the best measure of EmONC facility availability must therefore account for both geographic accessibility as well as facility readiness and evidence of functionality. Moreover, the population in need may not be distributed evenly so that in population-dense areas, travel-time may not be the only relevant measure of accessibility. Even with a sufficient number of facilities, appropriately distributed, socioeconomic dimensions of accessibility are important to take into account.

Finally, the dimensions of both "Acceptability" and "Quality" focus on the provision of appropriate, evidence-based care. As Gabrysch et al. have pointed out, the term "EmONC" conflates emergency care interventions with the presence of emergency facilities [16], thus measurement of this construct is not meaningful without a measure of facility functionality. Numerous studies show a significant deficit in functionality among facilities designated as providing EmONC. [40–42] Removing those facilities that are not functioning well affects the density and distribution of facilities but may not affect adequate availability of emergency care. There is a need to review the effectiveness and indication for these interventions based on current best available evidence and to examine data on their coverage and quality. Studies of EmONC facility performance of all signal functions required for the level of EmONC designated (basic or comprehensive) within the previous 90 days, suggest that a few signal functions (e.g., manual removal of products of conception and assisted vaginal delivery) are seldom evidenced in facilities that may otherwise be considered functional. [43,44] We found that very few facilities designated as emergency care facilities evidenced performance of all or most BEmONC or CEmONC signal functions, which suggests that using facility designation to calculate the numerator would over-estimate actual availability of EmoNC. Further, evidence of

specific elements of facility readiness to deliver all emergency services was also found to be lacking and, not surprisingly, was associated with a lack of EmONC signal function performance across the study sites. Facility readiness to deliver emergency interventions reliably and effectively, evidenced by the availability of essential drugs and core staff, is a core element of the construct of EmONC availability.

Systematically unpacking the various underlying components of the full construct for measurement reflecting EmONC sufficiency and exploring how they are reflected in various approaches to calculating the value of the estimate of Availability of EmONC in our study settings using multiple, innovative primary data and population data is a major strength of our study. Further, our facility sample consists of a census of all birthing sites in each study area, resulting in robust indicator estimates, which we believe is another strength. Our results are subject to some limitations as well. Given that the exact population in each of the study areas is unknown, we estimated the total population, population of women of reproductive age, and estimated births using publicly available data from WorldPop, using an analytical approach that has been published elsewhere [45]. Other limitations are primarily related to the scope of the research. For feasibility reasons, our study data are limited to specific subnational areas, not nationwide in study countries. Furthermore, we calculate travel time based on any mode of transportation; travel time estimates could be further examined by calculating travel time only by foot [26]. The study districts were selected to reflect a range of maternal health system performance, but characteristics of the study districts such as area and population density that are germane to the study question may vary across each country. Thus, our results may not be generalizable. Similarly, geographic calculation of two-hour travel time is limited to the population of the districts in our study sample and does not extend beyond district boundaries. Thus, a proportion of the population that lies outside the district may nevertheless be within the two-hour catchment area. To explore how changing these parameters would affect our results, we performed a sensitivity analysis of two districts in Ghana that border each other, allowing the facilities in Sunyani, a district that has several fully functioning EmONC facilities, to serve the population of Techiman, a district without any fully or partially functional EmONC facilities. Incorporating the facilities in Sunyani increases the percent of the total population in Techiman within two-hours travel time of a partially or fully functional EmONC facility from 0.0% to 95.6. Future research on indicators should consider how to handle accessibility across subnational borders, as accessibility across such borders may vary between settings. Finally, we had originally intended to include both pregnancies and births in our analysis; however, the differences in coverage estimates obtained using both denominators were so little as to not be meaningful. As a result, we only present data on births.

Future research should the explore the most effective array of evidence-based interventions based on epidemiology of risk that should constitute EmONC signal functions. More evidence is also needed to explore the optimal distribution of facilities and organization of care services to provide both effective routine care for physiologic pregnancy and childbirth and emergency care for complications to address all causes of maternal mortality, reproductive and maternal morbidity, and related disabilities, a key theme for ending preventable maternal mortality.

## Conclusions

Our study highlights significant differences in the value of estimates of sufficient EmONC coverage derived from country data depending on the definition of the indicator and measurement approach used. The optimal definition and calculation of a core measure to capture this construct is subject to uncertainty and the global reference standard indicators are currently under revision. Our study applies primary data to generate evidence that can help inform the

debate. To provide a valid measure of effective coverage of EmONC, future indicators, such as those included in global guidelines emitted by WHO, should go beyond the most narrow definition of availability of emergency facilities to include dimensions of AAAQ, including evidence of regular performance of emergency signal functions, facility readiness to do so reliably and effectively, and appropriate geographic distribution for accessibility to functional facilities by the best representation of the population in need.

## Supporting information

**S1 Checklist. Inclusivity in global research checklist.**
(DOCX)

## Acknowledgments

The authors would like to thank the following people, without whose efforts the publication of this manuscript would not have been possible:

In Argentina, we gratefully acknowledge the support of the National Directorate of Maternal, Child and Adolescent Health and the Directorate of Sexual and Reproductive Health of the Ministry of Health of the Nation. We commend the commitment and dedication of the provincial teams, and the following members of the Maternal and Child Health Programs of the Provincial Ministries of Health: Dr. Adriana Martirena, Dr. Daniel Nowacky, Dr. Adriana Allones, Marta Ferrary, Dr. Claudia Castro, Ana Seimande, Antonio Tabarcachi, Noelia Coria, Cintia Jacobi, Laura Soto, Dr. Mara Bazán, Dr. Susana Velazco, Dr. Patricia Leal, and Marcela Tapia. Finally, we would like to express our deepest gratitude to all of the health workers who participated in the study as data collectors, working through the height of the COVID-19 pandemic in Argentina.

In Ghana, we gratefully acknowledge the support of the Ghana Health Service Family Health Division, The Director General–Ghana Health Service–Dr. Patrick Kuma-Aboagye; Dr. Ernest Konadu Asiedu, Ms. Roberta Asiedu, Dr. Margretta Chandi and Ms. Catherine Adu Asare; Dr. Benedicta Mensah, Ms. Keziah Dampare, and all regional and district health workers and field teams for their persistence in data collection despite the challenges.

In India, we gratefully acknowledge the support of Dr. Dinesh Baswal, Ex Deputy Commissioner at Maternal Health Division, Ministry of Health & Family Welfare, India; the Mission Directors, State Health Departments of Tamil Nadu and Uttar Pradesh, and the District health Officials of study districts. We also acknowledge the support of Dr. Manju Chhugani and Dr. Renu Kharb for their guidance in review of the secondary data on many indicators. Finally, we sincerely thank the district field teams for their untiring efforts and adaptation to new methodologies to collect good quality data, in the midst of COVID in India. We also thank all the health workers and facility staff who participated in the study despite their busy schedules due to COVID situation.

Finally, we acknowledge Jean-Pierre Monet at UNFPA for his consultation in the formative stages of methodology development.

## Author Contributions

**Conceptualization:** Jewel Gausman, Verónica Pingray, Richard Adanu, Mabel Berrueta, Jeff Blossom, Suchandrima Chakraborty, Ernest Kenu, Ana Langer, Sowmya Ramesh, Niranjan Saggurti, Paula Vázquez, R. Rima Jolivet.

**Data curation:** Delia A. B. Bandoh, Suchandrima Chakraborty, Nizamuddin Khan, Magdalene A. Odikro, Sowmya Ramesh, Caitlin R. Williams.

**Formal analysis:** Jewel Gausman, Suchandrima Chakraborty, Winfred Dotse-Gborgbortsi, Nizamuddin Khan, Carolina Nigri, Magdalene A. Odikro, Sowmya Ramesh, Caitlin R. Williams.

**Funding acquisition:** Ana Langer, R. Rima Jolivet.

**Methodology:** Jewel Gausman, Verónica Pingray, Richard Adanu, Delia A. B. Bandoh, Mabel Berrueta, Jeff Blossom, Ernest Kenu, Magdalene A. Odikro, Sowmya Ramesh, Niranjan Saggurti, Paula Vázquez, R. Rima Jolivet.

**Project administration:** Jewel Gausman, Verónica Pingray, Delia A. B. Bandoh, Mabel Berrueta, Suchandrima Chakraborty, Ernest Kenu, Niranjan Saggurti, Paula Vázquez, R. Rima Jolivet.

**Supervision:** Jewel Gausman, R. Rima Jolivet.

**Validation:** Jeff Blossom, Winfred Dotse-Gborgbortsi.

**Visualization:** Jewel Gausman, Jeff Blossom.

**Writing – original draft:** Jewel Gausman, Jeff Blossom.

**Writing – review & editing:** Jewel Gausman, Verónica Pingray, Richard Adanu, Delia A. B. Bandoh, Mabel Berrueta, Suchandrima Chakraborty, Ernest Kenu, Nizamuddin Khan, Ana Langer, Carolina Nigri, Magdalene A. Odikro, Sowmya Ramesh, Niranjan Saggurti, Paula Vázquez, Caitlin R. Williams, R. Rima Jolivet.

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
