## [Decision Letter · Decision Letter 0]

21 Mar 2023

PONE-D-23-01594Validating indicators for monitoring availability and geographic distribution of emergency obstetric and newborn care (EmoNC) facilities: A study triangulating health system, facility, and geospatial dataPLOS ONE

Dear Dr. Gausman,

Thank you for submitting your manuscript to PLOS ONE. After careful consideration, we feel that it has merit but does not fully meet PLOS ONE’s publication criteria as it currently stands. Therefore, we invite you to submit a revised version of the manuscript that addresses the points raised during the review process.

We look forward to receiving your revised manuscript.

Kind regards,

Kyaw Lwin Show, MPH

Academic Editor

PLOS ONE

3. Please include a complete copy of PLOS’ questionnaire on inclusivity in global research in your revised manuscript. Our policy for research in this area aims to improve transparency in the reporting of research performed outside of researchers’ own country or community. The policy applies to researchers who have travelled to a different country to conduct research, research with Indigenous populations or their lands, and research on cultural artefacts. The questionnaire can also be requested at the journal’s discretion for any other submissions, even if these conditions are not met.  Please find more information on the policy and a link to download a blank copy of the questionnaire here: https://journals.plos.org/plosone/s/best-practices-in-research-reporting. Please upload a completed version of your questionnaire as Supporting Information when you resubmit your manuscript.

5. We note that Figures 3,4 and 6 in your submission contain [map/satellite] images which may be copyrighted. All PLOS content is published under the Creative Commons Attribution License (CC BY 4.0), which means that the manuscript, images, and Supporting Information files will be freely available online, and any third party is permitted to access, download, copy, distribute, and use these materials in any way, even commercially, with proper attribution. For these reasons, we cannot publish previously copyrighted maps or satellite images created using proprietary data, such as Google software (Google Maps, Street View, and Earth). For more information, see our copyright guidelines: http://journals.plos.org/plosone/s/licenses-and-copyright.

a. You may seek permission from the original copyright holder of Figures 3,4 and 6 to publish the content specifically under the CC BY 4.0 license. 

Reviewers' comments:

Reviewer's Responses to Questions

**Comments to the Author**

1. Is the manuscript technically sound, and do the data support the conclusions?

Reviewer #1: Yes

Reviewer #2: Yes

2. Has the statistical analysis been performed appropriately and rigorously? 

Reviewer #1: Yes

Reviewer #2: N/A

3. Have the authors made all data underlying the findings in their manuscript fully available?

Reviewer #1: Yes

Reviewer #2: Yes

4. Is the manuscript presented in an intelligible fashion and written in standard English?

Reviewer #1: Yes

Reviewer #2: Yes

5. Review Comments to the Author

Reviewer #1: This is an interesting study that seeks to validate

“Availability of EmONC” by comparing the value of the indicator after accounting for key aspects of

facility functionality and an alternative measure of geographic distribution.

There are a number of issues.

1. There appears to be some assumptions in the calculations. Example: The authors stated: Population estimates were generated using ArcGIS software using WorldPop to

estimate the total population, and the number of women of reproductive age (WRA), pregnancies and

births in the study areas. This need to be reflected in the limitation of the study.

2. The authors used several abbreviations that were not defined first before its use. For example, BEmONC and CEmONC. Also EPMM, AAAQ

3. Why are there non equal facilities recruited in the studied countries: example: The authors stated: 34 birthing facilities in Argentina, 51 in Ghana, and 282 in India

4. The introduction should end with the statement stating the overall objectives of the study.

5. The authors should stste how the sample size of the study was determined.

6. There were some mix up in the discussion section. The authors stated: To explore how changing these parameters would

affect our results, we performed a sensitivity analysis of two districts in Ghana that border each other,

allowing the facilities in Sunyani, a district that has several fully functioning EmONC facilities, to serve

the population of Techiman, a district without any fully or partially functional EmONC facilities.

Incorporating the facilities in Sunyani increases the percent of the total population in Techiman within

two-hours travel time of a partially or fully functional EmONC facility from 0.0% to 95.6%, as shown in

Figure 6. Future research ....

Figure 6: Geographic distribution of population within two hours travel time of a partially functional

EmONC facility allowing for the population in Techiman North to access facilities in Sunyani Municipal

in Ghana.

These confusion in the discussion section should be deleted.

7. The authors should discuss the streths and limitations of the study.

Reviewer #2: Thank you for allowing to review this paper which is very interesting. This paper highlights to consider the indicators for expanding to reflect the situation of EmONC in terms of all WHO AAAQ dimensions. While it is very important area to reduce MMR, I would like to suggest the following points;

- It should be clearly elaborated the main findings and key proposed indicators to expand/ add to be more comprehensive reflection the status on EmONC in addition to current WHO indicators in conclusion part. That can be easily picked up by readers.

-In results, the interpretation for each tables should be more clear/convinced to readers. Now, there are many figures and difficult to follow.

-In table 2 and 3, total (100%) may be row percent; should be column percent to be more informative.

-In table 4, While there is no assigned facilities, there are some records for that signal functions. What is the meaning of showing p value; and the meaning of significant of p value should be elaborated in text. What does mean the association between assigned facilities and records on signal function? It is not clear.

-Overall, there are many tables and figures, but it should be more clear to readers what are the key messages.

6. PLOS authors have the option to publish the peer review history of their article (what does this mean?). If published, this will include your full peer review and any attached files.

Reviewer #1: **Yes: **George Eleje

Reviewer #2: No

---

## [Author Response · Author response to Decision Letter 0]

25 May 2023

Response: We have adjusted the formatting of our manuscript and cover page to conform to PLOS ONE’s style requirements, detailed above. 

Response: We have added the sentence below under the subsection “Ethics statement.” We have removed the ethics statement under declarations as instructed. 

“The Institutional Review Board (IRB) of the Harvard T.H. Chan School of Public Health approved this study on 4 September 2019 (approval ID: IRB19-1086). The research is classified as Level 4 Data using Harvard’s Data Security Policy. The study also was approved in Argentina by the Comité de Ética de la Investigación de la Provincia de Jujuy (approval ID not applicable), Comisión Provincial de Investigaciones Biomédicas de la Provincia de Salta (approval ID: 321-284616/2019), Consejo Provincial de Bioética de la Provincia de La Pampa (approval ID not applicable), and Comité de Ética Central de la Provincia de Buenos Aires (approval ID: 2919-2056-2019); in Ghana by the Ghana Health Service Ethical Review Board (approval ID: GHS-ERC022/08/19); and in India by the national population council IRB (approval ID: 889) and local Sigma-IRB (approval ID: 10052/IRB/19-20).

After full board review, the need for informed consent was waived as it was determined our study did not collect any data on human subjects. Only anonymized and generalizable data were collected during retrospective medical chart review.”

3. Please include a complete copy of PLOS’ questionnaire on inclusivity in global research in your revised manuscript. Our policy for research in this area aims to improve transparency in the reporting of research performed outside of researchers’ own country or community. The policy applies to researchers who have travelled to a different country to conduct research, research with Indigenous populations or their lands, and research on cultural artefacts. The questionnaire can also be requested at the journal’s discretion for any other submissions, even if these conditions are not met. Please find more information on the policy and a link to download a blank copy of the questionnaire here: https://journals.plos.org/plosone/s/best-practices-in-research-reporting. Please upload a completed version of your questionnaire as Supporting Information when you resubmit your manuscript.

Response: As suggested above, we have completed the “Inclusivity in global research” questionnaire and included it as supplementary information. We have also included a subsection in the “Methods and materials” section of the paper as follows and suggested: 

“Inclusivity in global research 

Additional information regarding the ethical, cultural, and scientific considerations specific to inclusivity in global research is included in the Supporting Information (S1 Checklist)”

Response: As noted above, we have removed the ethics statement from under the declarations section and included it in “Methods and materials.”

5. We note that Figures 3,4 and 6 in your submission contain [map/satellite] images which may be copyrighted. All PLOS content is published under the Creative Commons Attribution License (CC BY 4.0), which means that the manuscript, images, and Supporting Information files will be freely available online, and any third party is permitted to access, download, copy, distribute, and use these materials in any way, even commercially, with proper attribution. For these reasons, we cannot publish previously copyrighted maps or satellite images created using proprietary data, such as Google software (Google Maps, Street View, and Earth). For more information, see our copyright guidelines: http://journals.plos.org/plosone/s/licenses-and-copyright.

a. You may seek permission from the original copyright holder of Figures 3,4 and 6 to publish the content specifically under the CC BY 4.0 license. 

Response: Thank you for noticing the background images on Figures 3,4,6. These figures do not contain any satellite background. The background layer being displayed is population density data from WorldPop, an open-source data source detailed in our paper. To further clarify this, we have added a new section in the “Methods and materials” section of the paper.

“Map Background Data Source

For the maps in Figures 3, 4, and 6 the “Population density” data displayed in the background was accessed from WorldPop (www.worldpop.org) and is available for use under the Creative Commons Attribution 4.0 International License (Argentina: https://hub.worldpop.org/doi/10.5258/SOTON/WP00674, Ghana: https://hub.worldpop.org/doi/10.5258/SOTON/WP00674, India: https://hub.worldpop.org/doi/10.5258/SOTON/WP00674).

Reviewers' comments:

Reviewer's Responses to Questions

Comments to the Author

1. Is the manuscript technically sound, and do the data support the conclusions?

Reviewer #1: Yes

Reviewer #2: Yes

2. Has the statistical analysis been performed appropriately and rigorously? 

Reviewer #1: Yes

Reviewer #2: N/A

3. Have the authors made all data underlying the findings in their manuscript fully available?

Reviewer #1: Yes

Reviewer #2: Yes

4. Is the manuscript presented in an intelligible fashion and written in standard English?

Reviewer #1: Yes

Reviewer #2: Yes

5. Review Comments to the Author

Reviewer #1: This is an interesting study that seeks to validate

“Availability of EmONC” by comparing the value of the indicator after accounting for key aspects of facility functionality and an alternative measure of geographic distribution.

There are a number of issues.

1. There appears to be some assumptions in the calculations. Example: The authors stated: Population estimates were generated using ArcGIS software using WorldPop to

estimate the total population, and the number of women of reproductive age (WRA), pregnancies and births in the study areas. This need to be reflected in the limitation of the study.

Response: Given that the exact population in each of the study areas is unknown, we estimated the total population, population of women of reproductive age, and estimated births using publicly available data from WorldPop, using an analytical approach that has been published elsewhere (45).

2. The authors used several abbreviations that were not defined first before its use. For example, BEmONC and CEmONC. Also EPMM, AAAQ

Response: We thank the reviewer for their detailed review of our manuscript; however, we note that both BEmONC and CEmONc were defined at their first use in the second paragraph on page 2. EPMM (referring to Ending Preventable Maternal Mortality) was defined in the first paragraph on page 1. AAAQ (referring the Availability, Accessibility, Acceptability, and Quality Framework) was defined at the first use of the term in the second paragraph on page 6. 

3. Why are there non equal facilities recruited in the studied countries: example: The authors stated: 34 birthing facilities in Argentina, 51 in Ghana, and 282 in India

Response: We thank the reviewer for this comment. As stated in the first sentence in the “Materials and methods” section, we included a census of all birthing facilities in each of the four subnational geographic areas in Argentina, Ghana, and India. Thus, the unequal numbers are explained by the fact that there are simply many more birthing sites in India than in the other two countries. No birthing facilities were omitted from the study. 

4. The introduction should end with the statement stating the overall objectives of the study.

Response: We have revised the final paragraph in the introduction to more clearly reference the aims of the study. The following sentences are meant to clarify our study aims: “This study aims to examine these first two dimensions on the impact on the value of the indicator if the numerator is redefined to account for facility performance and functionality, and also if the denominator is redefined to reflect different population groups that are considered to be in need (total population, women of reproductive age, and births),” and “This study aims to explore this dimension by estimating the proportion the population within two-hours travel time.”

5. The authors should stste how the sample size of the study was determined.

Response: We thank the reviewer for this comment. The first sentence under “Materials and methods” states that we included a census of all birthing facilities in each of the four subnational geographic areas in Argentina, Ghana, and India. As we included all birthing facilities (i.e., a census), there is no sample size calculation or determination. 

6. There were some mix up in the discussion section. The authors stated: To explore how changing these parameters would affect our results, we performed a sensitivity analysis of two districts in Ghana that border each other, allowing the facilities in Sunyani, a district that has several fully functioning EmONC facilities, to serve the population of Techiman, a district without any fully or partially functional EmONC facilities.

Incorporating the facilities in Sunyani increases the percent of the total population in Techiman within two-hours travel time of a partially or fully functional EmONC facility from 0.0% to 95.6%, as shown in Figure 6. Future research ....

Figure 6: Geographic distribution of population within two hours travel time of a partially functional EmONC facility allowing for the population in Techiman North to access facilities in Sunyani Municipal in Ghana. These confusion in the discussion section should be deleted.

Response: We have removed Fig 6 from the paper.

7. The authors should discuss the streths and limitations of the study.

Response: We kindly refer the reviewer to the second paragraph on page 38 in which we discuss both the strengths and limitations of the study. 

Reviewer #2: Thank you for allowing to review this paper which is very interesting. This paper highlights to consider the indicators for expanding to reflect the situation of EmONC in terms of all WHO AAAQ dimensions. While it is very important area to reduce MMR, I would like to suggest the following points;

- It should be clearly elaborated the main findings and key proposed indicators to expand/ add to be more comprehensive reflection the status on EmONC in addition to current WHO indicators in conclusion part. That can be easily picked up by readers.

Response: We thank the reviewer for this comment. While suggesting new indicators is beyond the scope of this study as this is a validation study, we suggest specific considerations that global bodies, such as WHO, may consider when revising indicators in the future. We have revised the last sentence of the manuscript to read as follows. “To provide a valid measure of effective coverage of EmONC, future indicators, such as those included in global guidelines emitted by WHO, should go beyond the most narrow definition of availability of emergency facilities to include dimensions of AAAQ, including evidence of regular performance of emergency signal functions, facility readiness to do so reliably and effectively, and appropriate geographic distribution for accessibility to functional facilities by the best representation of the population in need.”

-In results, the interpretation for each tables should be more clear/convinced to readers. Now, there are many figures and difficult to follow.

Response: We thank the reviewer for this comment. We believe that each of our tables and figures display important elements of our validation study, which is admittedly complex. To respond to the reviewers’ comment, we have included a one-sentence, plain language, overall summary of each table and figure in order to make them easier for the reader to interpret (except for Table 1, which we believe is straightforward). We have also removed Fig 6 in order to reduce the number of figures. 

Table 2: “As shown in the table, the vast majority of BEmONC-designed facilities in Ghana and India were found to be non-performing.”

Table 3: “While performance of EmONC signal functions at CEmONC-designated facilities was better than among BEmONC-designated facilities, a substantial proportion of CEmONC-designated facilities did not perform all nine CEmONC signal functions or all seven BEmONC signal functions in all three countries.”

Table 4: “In other words, facilities that did not perform a specific signal function may not have had an obstetric emergency that necessitated its performance during the 90 days prior to data collection.” 

Table 5: “In general, a majority of birthing facilities in all three countries were open to provide 24/7 emergency care, had at least one staff on call or on duty 24/7 to respond to an obstetric emergency, had all essential drugs in stock (except for among non-EmONC designated facilities in Ghana), and had a drug inventory register present on site (Table 5).”

Table 6: “After having examined the different dimensions that comprise the standard and alternative indicator definitions, we now calculate the number of facilities per 500,000 total population and 20,000 births, while accounting for facility designation and functionality (Table 6).”

Table 7: “The percentage of the total population, women of reproductive age, and births occurring within two hours travel time of a BEmONC-designated, partially functioning BEmONC, or fully functioning BEmONC facility is presented in Table 7. Changing the population parameter between total population, women of reproductive age, and births, produces little variation across sub-national study areas, and in most areas, does not affect whether the coverage target of 90% of the population within two-hours travel-time is met. However, there are some notable coverage reductions observed when changing the population parameters.” (note: this is unchanged, we believe that this is sufficient interpretation). 

Fig 1: “In general, across all three countries, there was a high degree of variability in the performance of each of the nine signal functions in birthing sites.”

Fig 2: “Further, the results are disaggregated by facility location (rural/urban) and level (primary, secondary, and tertiary) to explore disparities by these characteristics. Across all countries, the vast majority of BEmONC-designated facilities, as well as facilities without EmONC designation, were non-functional. While in general, a larger proportion of CEmONC-designated displayed some degree of functionality, an important percentage were non-functional.”

Figs 3 and 4: “According to the figures, the location of partially or fully functional facilities does not necessarily correlate with areas have the highest population density within a district, and further, in many districts, even if there is only one partially or fully functional EmONC facility, the majority of the population remains within two-hours travel time to the facility.”

Fig 5: “In general, changing the numerator or denominator leads to indicator estimates that exceed the suggested targets in one domain but not the other, thus leading to conflicting interpretations of adequacy.”

Fig 6: We have removed Fig 6. 

-In table 2 and 3, total (100%) may be row percent; should be column percent to be more informative.

Response: We thank the reviewer for their careful review, however, we disagree with the reviewers’ assessment of row versus column percents. In both of the tables in question, we are interested in knowing the distribution of EmONC performance within each strata. For example, we want to know what proportion of BEmONC-designated facilities within Bunkpurugu Yunyoo exhibited full BEmONC performance, partial BEmONC performance, or were considered to be non-performing. Looking at the row totals in this sense gives us an idea about facility performance among facilities within each district, level within the health system (type), and location. Conversely, column totals, as suggested by the reviewer, would give us the distribution of BEmONC-designated facilities across these stratifying variables. Following the same example, knowing the proportion of BEmONC facilities that performed all signal functions that exist in Bunkpurugu Yunyoo would not give us insight into how facilities within that district are performing. 

-In table 4, While there is no assigned facilities, there are some records for that signal functions. What is the meaning of showing p value; and the meaning of significant of p value should be elaborated in text. What does mean the association between assigned facilities and records on signal function? It is not clear.

Response: In the methods section (on page 13) we have added an explanation of why we compare case load of obstetric emergencies to signal function performance. “As a final examination of the numerator’s indicator, we used chi squared tests to explore whether facility performance of specific signal functions over the previous 90 days was related to whether the facility had at least one case of an obstetric emergency that would warrant the performance of that specific signal function. As some clinics may not encounter the specific obstetric emergencies that may warrant performance of a given signal function, this enables us to examine whether lack of performance of a signal function may be explained by case load, rather than a lack of ability to perform that signal function.”

We have also revised the paragraph in the results explaining Table 4 to read as follows: 

“A comparison of a facility’s performance of a specific signal function during the 90 days prior to data collection to whether a facility had record of a corresponding obstetric emergency during that same time period suggests that in general, there is a significant association between case load and performance, as detailed in Table 4. For the most part, facilities that did not perform a specific signal function had no record of encountering an obstetric emergency that would require its performance. For example, among the facilities that failed to perform Signal Function #3 (administration of anticonvulsants), more than 80% did not have a confirmed case of severe pre-eclampsia in the facility register which would be a clinical reason to administer anticonvulsants. There are some important exceptions to note. In India, a majority (61.1%) of facilities that did not perform Signal Function #9 (blood transfusion) had at least one case of maternal hemorrhage on file.”

-Overall, there are many tables and figures, but it should be more clear to readers what are the key messages.

Response: We believe that we have addressed this comment in response to the reviewer’s previous comments. 

6. PLOS authors have the option to publish the peer review history of their article (what does this mean?). If published, this will include your full peer review and any attached files.

Do you want your identity to be public for this peer review? For information about this choice, including consent withdrawal, please see our Privacy Policy.

Reviewer #1: Yes: George Eleje

Reviewer #2: No

---

## [Decision Letter · Decision Letter 1]

15 Jun 2023

Validating indicators for monitoring availability and geographic distribution of emergency obstetric and newborn care (EmoNC) facilities: A study triangulating health system, facility, and geospatial data

PONE-D-23-01594R1

Dear Dr. Gausman,

We’re pleased to inform you that your manuscript has been judged scientifically suitable for publication and will be formally accepted for publication once it meets all outstanding technical requirements.

Kind regards,

Kyaw Lwin Show, MPH

Academic Editor

PLOS ONE

Additional Editor Comments (optional):

Reviewers' comments:

Reviewer's Responses to Questions

**Comments to the Author**

1. If the authors have adequately addressed your comments raised in a previous round of review and you feel that this manuscript is now acceptable for publication, you may indicate that here to bypass the “Comments to the Author” section, enter your conflict of interest statement in the “Confidential to Editor” section, and submit your "Accept" recommendation.

Reviewer #2: All comments have been addressed

2. Is the manuscript technically sound, and do the data support the conclusions?

Reviewer #2: Yes

3. Has the statistical analysis been performed appropriately and rigorously? 

Reviewer #2: Yes

4. Have the authors made all data underlying the findings in their manuscript fully available?

Reviewer #2: Yes

5. Is the manuscript presented in an intelligible fashion and written in standard English?

Reviewer #2: Yes

6. Review Comments to the Author

Reviewer #2: Thank you for sharing the revised manuscripts. The authors have addressed all the comments. I have no additional comment.

7. PLOS authors have the option to publish the peer review history of their article (what does this mean?). If published, this will include your full peer review and any attached files.

Reviewer #2: No

---

## [Editor Report · Acceptance letter]

6 Sep 2023

PONE-D-23-01594R1 

Validating indicators for monitoring availability and geographic distribution of emergency obstetric and newborn care (EmoNC) facilities: A study triangulating health system, facility, and geospatial data 

Dear Dr. Gausman:

I'm pleased to inform you that your manuscript has been deemed suitable for publication in PLOS ONE. Congratulations! Your manuscript is now with our production department. 

Kind regards, 

on behalf of

Dr. Kyaw Lwin Show 

Academic Editor

PLOS ONE